# Unlocking the Power of Multi-Agent LLM for Reasoning: From Lazy Agents to Deliberation

**Zhiwei Zhang**[1], **Xiaomin Li**[2], **Yudi Lin**[3], **Hui Liu**[4], **Ramraj Chandradevan**[4], **Linlin Wu**[5], **Minhua Lin**[1], **Fali Wang**[1], **Xianfeng Tang**[4], **Qi He**[6], **Suhang Wang**[1]*

[1]The Pennsylvania State University  [2]Harvard University  [3]Michigan State University
[4]Amazon  [5]University of Utah  [6]Microsoft
{zbz5349, szw494}@psu.edu

## ABSTRACT

Large Language Models (LLMs) trained with reinforcement learning and verifiable rewards have achieved strong results on complex reasoning tasks. Recent work extends this paradigm to a multi-agent setting, where a meta-thinking agent proposes plans and monitors progress while a reasoning agent executes subtasks through sequential conversational turns. Despite promising performance, we identify a critical limitation: lazy agent behavior, in which one agent dominates while the other contributes little, undermining collaboration and collapsing the setup to an ineffective single agent. In this paper, we first provide a theoretical analysis showing why lazy behavior naturally arises in multi-agent reasoning. We then introduce a stable and efficient method for measuring causal influence, helping mitigate this issue. Finally, as collaboration intensifies, the reasoning agent risks getting lost in multi-turn interactions and trapped by previous noisy responses. To counter this, we propose a verifiable reward mechanism that encourages deliberation by allowing the reasoning agent to discard noisy outputs, consolidate instructions, and restart its reasoning process when necessary. Extensive experiments demonstrate that our framework alleviates lazy agent behavior and unlocks the full potential of multi-agent framework for complex reasoning tasks.

## 1 INTRODUCTION

Recent advances in prompting and training have markedly improved the multi-step reasoning abilities of large language models (LLMs) (Wei et al., 2022; Kojima et al., 2022; Wang et al., 2022; Zhang et al., 2022; Ton et al., 2024; Yeo et al., 2025; Zhu et al., 2025; Chowdhury & Caragea, 2025; Mukherjee et al., 2025; Balcan et al., 2025). Techniques such as chain-of-thought prompting (Wei et al., 2022; Kojima et al., 2022) and structured methods like Tree-of-Thoughts and Graph-of-Thoughts (Yao et al., 2023; Besta et al., 2024) expand the space for deliberation. Building on this, Large Reasoning Models (LRMs) trained with supervised and reinforcement learning using verifiable rewards achieve strong performance on math, code, and planning tasks (Jaech et al., 2024; Guo et al., 2025a; Comanici et al., 2025; Plaat et al., 2024; Huang & Chang, 2022; Zhang et al., 2023b; Li et al., 2025b; Chen et al., 2025). More recently, multi-agent frameworks enable LLMs with specialized roles to collaborate via planning, delegation, and debate, echoing human team dynamics (Li et al., 2023; Wu et al., 2024a; Chen et al., 2023; Du et al., 2023; Yuan & Xie). Likewise, single-agent multi-turn interaction settings have gained attention as another path to enhance reasoning (Wan et al., 2025; Shi et al., 2025; Wei et al., 2025; Zhou et al., 2025; Li et al., 2025c; Lu et al., 2025; Wang et al., 2025; Zhang et al., 2025a; Zeng et al., 2025; Jin et al., 2025).

To support multi-agent and multi-turn reinforcement learning, multi-turn Group Relative Policy Optimization (GRPO) (Wan et al., 2025; Shi et al., 2025; Wei et al., 2025) and its variants (Guo et al., 2025b; Zhang et al., 2025c; Ning et al., 2025; Xue et al., 2025) compute advantages and importance ratios at the turn level, enabling finer-grained optimization and more precise credit assignment. Building on this foundation, ReMA (Wan et al., 2025) introduces a multi-agent LLM reasoning

---

*Corresponding Author.

framework with two specialized roles: a *meta-thinking agent*, which decomposes tasks, sets intermediate goals, and adapts based on feedback, and a *reasoning agent*, which performs step-by-step computations and proofs before returning intermediate results. The agents alternate sequentially, but since only a final outcome reward is available, ReMA computes a group advantage following GRPO (Shao et al., 2024) and uniformly assigns this trajectory-level signal to every turn in the rollout.

Despite its effectiveness, we empirically find that ReMA suffers from a critical issue of lazy agents: one of the agents contributes only trivially to the multi-agent system. Although this phenomenon has been widely acknowledged in traditional multi-agent reinforcement learning under sparse-reward settings (Sunehag et al., 2018; Foerster et al., 2018; Castellini et al., 2022; Jaques et al., 2019; Wang et al., 2020; Liu et al., 2023), prior work has primarily focused on scenarios where multiple agents act simultaneously. *In contrast, our findings are surprising because agents in our setting act sequentially.* An early agent's trivial action not only fails to contribute but also shapes the state for subsequent agents. As later decisions depend on this evolving state, a lazy action can misguide the reasoning trajectory and compound its negative influence. Intuitively, such interdependence across turns should discourage laziness, especially as overall performance improves during training. However, contrary to this expectation, we find that ReMA-trained reasoning agents still adopt shortcut behaviors. As shown in Section 4, our case study reveals that reasoning agents often contribute only trivially, typically by summarizing or copying the meta-thinking agent's responses without genuine questioning or reflection. As a result, the meta-thinking agent ends up carrying almost the entire reasoning process. Our causal-effect experiments further show that while both agents initially contribute substantially when initialized from the base model, the reasoning agent's influence diminishes markedly as training progresses, leaving the meta-thinking agent dominant.

The critical issue of lazy agents in multi-agent systems risks collapsing the entire system into a single agent, thereby limiting the potential benefits of collaboration in improving performance. In this paper, we propose Multi-Agent Meta-Reasoning Done Right (**Dr. MAMR**). We begin with a theoretical analysis of multi-turn GRPO to investigate the root cause of lazy agent behavior and identify a key bias in its loss formulation: the normalization term, intended to prevent sequence-level bias toward longer rollouts, inadvertently drives the model to prefer continuations that minimize the number of turns given the same prefix. As a result, agents are implicitly incentivized to complete reasoning with fewer interactions, often bypassing collaborative reflection or correction, and over time, this dynamic gives rise to lazy agents that contribute little to the reasoning process. Our theoretical insight not only explains the emergence of lazy agents but also sheds light on future work in designing objectives for multi-turn reinforcement learning.

While correcting the loss formulation partially mitigates the problem, it does not eliminate it. To further address this issue, we propose measuring the causal influence (Bogdan et al., 2025) of each reasoning step on subsequent process. A challenge arises in online training: the policy generates only a single continuation per step, so the estimated influence reflects just one trajectory. In contrast, considering multiple continuations would show how the step contributes across diverse trajectories, providing a more reliable estimate of its overall contribution and mitigating potential bias introduced by phrasing (Pavlick & Callison-Burch, 2016; McCoy et al., 2019; Merrick & Taly, 2020; Li et al., 2024), but such resampling is computationally prohibitive in online RL. Inspired by Feng et al. (2025); Li et al. (2021), we introduce a Shapley-inspired causal influence method. Instead of evaluating each step in isolation, we group semantically similar steps across rollouts and average their influence scores. This avoids additional sampling and produces robust estimates during training.

As lazy behavior diminishes and agents engage more productively, interaction frequency increases. However, as Laban et al. (2025) show, LLMs in multi-turn settings often overcommit to incomplete early context, making premature assumptions. A similar risk arises here: the meta-thinking agent acts like a user providing incremental instructions, while the reasoning agent may become misled by its own earlier outputs, as confirmed in Sec. 5.3. To overcome this, we propose training the reasoning agent to adaptively discard its prior outputs, re-aggregate instructions, and restart reasoning when needed. To accurately credit such restart behavior, we design a novel verifiable reward mechanism. Building on this, we assign step-level credit by aggregating outcome reward, causal influence, and restart signals. Extensive experiments demonstrate that our method effectively mitigates lazy-agent behavior and unlocks the potential of multi-agent frameworks for complex reasoning.

Overall, our contributions are as follows: **(1)** We identify a critical issue of lazy agents in multi-agent reasoning frameworks and provide a theoretical analysis of multi-turn GRPO to explain its

underlying cause. **(2)** We propose a Shapley-inspired method for measuring causal influence at the step level, further mitigating the lazy agent problem. **(3)** As agents engage in more frequent collaboration, we design a novel verifiable reward mechanism for restart behavior, enabling the reasoning agent to recover from noisy intermediate steps and avoid getting lost in prolonged interactions, thereby pushing the performance boundary of multi-agent LLMs on complex reasoning tasks.

## 2 RELATED WORK

**Multi-Agent RL.** Multi-agent RL (MARL) studies how agents coordinate to maximize collective rewards, with credit assignment as a central challenge. Classical approaches include value decomposition (Sunehag et al., 2018), counterfactual baselines (Foerster et al., 2018), regression-based rewards (Castellini et al., 2022), role-based coordination (Wang et al., 2020), and model-based influence estimation (Liu et al., 2023). With LLM agents, MARL has been adapted for multi-turn reasoning and dialogue, e.g., turn-level credit assignment (Zeng et al., 2025), critic-driven step-wise rewards (Zhou et al., 2025), communication-efficient training (Liao et al., 2025), addressing coarse reward traps (Wang et al., 2025), and framing LLM collaboration via MAGRPO (Liu et al., 2025a). A persistent issue is lazy agents, motivating causal influence estimation (Bogdan et al., 2025; Nguyen et al., 2025; Liu et al., 2024b) to enable finer-grained credit assignment.

**LLM reasoning.** Large Language Models (LLMs) excel across diverse NLP tasks (Brown et al., 2020; Chowdhery et al., 2023; Du et al., 2022; Dubey et al., 2024; Wenzek et al., 2019). Chain-of-thought prompting improves reasoning by eliciting intermediate steps (Wei et al., 2022; Kojima et al., 2022; Nye et al., 2021), while extensions like Tree-of-Thoughts and Graph-of-Thoughts enable structured, non-linear reasoning (Yao et al., 2023; Besta et al., 2024). These advances motivate Large Reasoning Models (LRMs) (Guo et al., 2025a; Achiam et al., 2023; Grattafiori et al., 2024; Xu et al., 2023; Zhou et al., 2022; Wu et al., 2024b; Qi et al., 2024; Chae et al., 2024), which combine supervised fine-tuning and reinforcement learning to achieve state-of-the-art results on math, coding, and planning (Jaech et al., 2024; Guo et al., 2025a; Comanici et al., 2025; Yang et al., 2024a; 2025; Lightman et al., 2023; Wang et al., 2023). Beyond single-model reasoners, multi-agent frameworks leverage role assignment, orchestration, and debate to coordinate specialized LLM agents for complex tasks (Li et al., 2023; Wu et al., 2024a; Chen et al., 2023; Du et al., 2023; Yuan & Xie).

A detailed review of related work on MARL, hierarchical RL and LLM reasoning is in Appendix A.

## 3 BACKGROUND

ReMA (Wan et al., 2025) models reasoning as a *multi-turn meta-thinking process* defined as:

$$\mathbf{x} \xrightarrow[\pi_h]{\text{meta-thinking}} \mathbf{m}_1 \xrightarrow[\pi_l]{\text{reasoning}} \mathbf{y}_1 \xrightarrow[\pi_h]{\text{meta-thinking}} \mathbf{m}_2 \xrightarrow[\pi_l]{\text{reasoning}} \mathbf{y}_2 \dots \xrightarrow[\pi_l]{\text{reasoning}} \mathbf{y}_T, \tag{1}$$

where $T$ is the number of turns. The high-level policy $\pi_h$ (*meta-thinking agent*) generates meta-level thoughts $\mathbf{m}_t$ from the input $\mathbf{x}$ and history $\{\mathbf{m}, \mathbf{y}\}_{<t}$, while the low-level policy $\pi_l$ (*reasoning agent*) produces token-level outputs $\mathbf{y}_t$ under the guidance of $\mathbf{m}_t$. To improve training efficiency, both agents share the same model weights $\theta$ but are distinguished by role-specific system prompts $S_h$ and $S_l$: $\pi_h = \pi_\theta(\cdot \mid S_h, \cdot), \pi_l = \pi_\theta(\cdot \mid S_l, \cdot)$.

**Multi-turn GRPO** (Wan et al., 2025; Wei et al., 2025; Shi et al., 2025) extends GRPO (Guo et al., 2025a) to support end-to-end multi-turn tasks such as mathematical reasoning (Wan et al., 2025) and web-based agent decision-making (Wei et al., 2025). A key innovation is the introduction of a turn-level importance ratio, enabling fine-grained credit assignment across dialogue turns.

Specifically, given the dataset $\mathcal{D}$ and $G$ trajectories for each question, the objective is defined as:

$$\mathcal{J}(\theta) = \mathbb{E}_{(\mathbf{x},\mathbf{y}^*)\sim\mathcal{D},\, \{(\mathbf{m}_i,\mathbf{y}_i)\}_{i=1}^{G}\sim\pi_{\theta_{\text{old}}}(\cdot|\mathbf{x})}$$

$$\left[ \frac{1}{G}\sum_{i=1}^{G}\frac{1}{T_i}\sum_{t=1}^{T_i}\frac{1}{|\mathbf{y}_{i,t}|}\sum_{j=1}^{|\mathbf{y}_{i,t}|}\left(\min\left(r_{i,t}(\theta)\hat{A}_{i,t,j},\, \text{clip}(r_{i,t}(\theta), 1-\epsilon, 1+\epsilon)\hat{A}_{i,t,j}\right) - \beta\, D_{\text{KL}}(\pi_\theta\|\pi_{\text{ref}})\right)\right], \tag{2}$$

where $\mathbf{y}_{i,t,j}$ denotes the $j$-th token at turn $t$ in trajectory $i$, $\hat{A}_{i,t,j}$ is the token-level advantage and $\frac{1}{T_i}$ is a normalization to avoid bias toward rollouts with more turns. The turn-level importance ratio

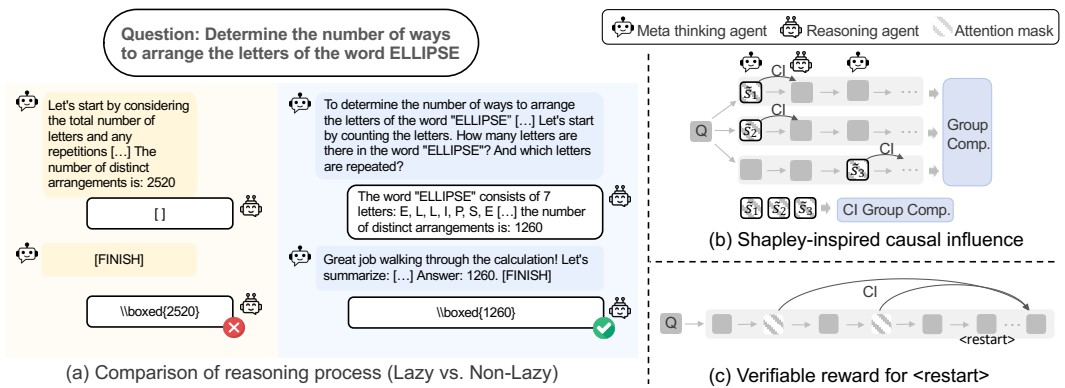

Figure 1: (a) Case study on lazy agents (full process in Appendix D); (b–c) our proposed modules.

$r_{i,t}(\theta)$ is computed as:

$$r_{i,t}(\theta) = \frac{1}{|\mathbf{y}_{i,t}|} \sum_{j=1}^{|\mathbf{y}_{i,t}|} \frac{\pi_\theta\big(\mathbf{y}_{i,t,j} \mid \mathbf{x}, \{\mathbf{m}_{i,\cdot}, \mathbf{y}_{i,\cdot}\}_{<t}, \mathbf{m}_{i,t}, \mathbf{y}_{i,t,<j}\big)}{\pi_{\theta_{\text{old}}}\big(\mathbf{y}_{i,t,j} \mid \mathbf{x}, \{\mathbf{m}_{i,\cdot}, \mathbf{y}_{i,\cdot}\}_{<t}, \mathbf{m}_{i,t}, \mathbf{y}_{i,t,<j}\big)}, \tag{3}$$

which aggregates token-level likelihood ratios within each reasoning turn. Similar variants of Eq. 2 were also proposed in (Guo et al., 2025b; Zhang et al., 2025c; Ning et al., 2025; Xue et al., 2025).

## 4 THE LAZY AGENT ISSUE IN MULTI-AGENT LLM REASONING

In this section, we present empirical evidence of the lazy-agent problem in the multi-agent framework ReMA (Wan et al., 2025). As illustrated in Fig. 1(a), the reasoning agent often outputs blanks at intermediate steps, shifting the burden to the meta-thinking agent and ultimately leading to incorrect answers. By contrast, when both agents actively contribute, collaboration yields correct solutions. To quantify laziness, we measure the causal influence of an agent's actions by adapting the attention-suppression method from Bogdan et al. (2025). Let $s_t$ denote an action taken by either agent. We suppress all attention (across layers and heads) to the tokens corresponding to $s_t$ and define the influence on the subsequent action $s_{t+1}$ as the KL divergence between the model's logits with and without suppression. Intuitively, a small divergence indicates that the agent's step has little impact on subsequent reasoning and thus reflects lazy behavior, whereas a large divergence shows that the step substantially shapes the reasoning process. See Appendix C.1 for experimental details.

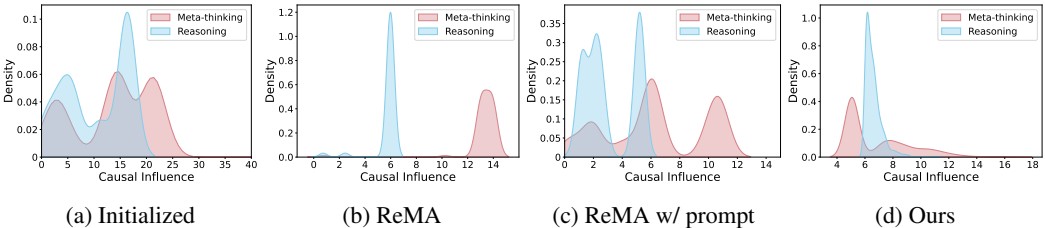

(a) Initialized      (b) ReMA      (c) ReMA w/ prompt      (d) Ours

Figure 2: Causal effect comparison. Performance on MATH500 under different configurations: (a) 75.0, (b) 74.4, (c) 75.6, and (d) 78.4.

To examine whether the lazy-agent issue arises from the ReMA framework, we compare causal influence under three settings: (1) untrained agents initialized from the base model, (2) agents trained with ReMA, and (3) agents trained with ReMA but prompted with instructions discouraging trivial responses (details in Appendix C.1). The results are shown in Fig. 2(a–c). **(1)** Compared to the untrained baseline, the reasoning agent in the standard ReMA setting contributes substantially less than the meta-thinking agent, revealing clear lazy-agent behavior. This imbalance coincides with a performance drop from 75.0 to 74.4 on MATH500 despite training. **(2)** Adding a prompt to encourage non-trivial responses narrows the causal-effect gap and improves performance from 74.4

to 75.6. However, the reasoning agent still shows weaker influence than in the baseline, and the modest 0.6-point gain suggests reliance on shortcuts rather than meaningful reasoning. In summary, ReMA is prone to producing lazy agents. While prompt engineering can partially mitigate the issue, it does not fully resolve it. This underscores the need for more robust methods, particularly in online reinforcement learning, to ensure balanced agent contributions.

# 5 DR. MAMR: MULTI-AGENT META-REASONING DONE RIGHT

## 5.1 THEORETICAL ANALYSIS ON THE EMERGENCE OF LAZY AGENT

Our preliminary experiments in Sec. 4 reveal the critical issue of lazy agents in advanced multi-agent reasoning frameworks. In this section, we provide a theoretical analysis of why such behavior emerges, even when the overall system performance appears to improve. Following GRPO (Shao et al., 2024) and DAPO (Yu et al., 2025), multi-turn GRPO (Wan et al., 2025; Wei et al., 2025) introduces a normalization term to reduce sequence-level bias toward longer rollouts. As shown in Eq. 2, the objective includes a factor $\frac{1}{T_i}$ that averages the turn-level advantages across each trajectory. However, we find that this normalization introduces a structural bias: *given the same context, if two alternative actions produce trajectories with equal final reward but different numbers of turns, the model favors the action leading to fewer turns*. To illustrate, we consider the reasoning agent as an example and present the following theory:

**Theorem 1** *Let $g_t(\tau) = \frac{1}{T(\tau)} Z_t(\tau)$ be the gradient contribution at turn $t$ for trajectory $\tau$ with $Z_t(\tau) \triangleq \frac{1}{|y_t|} \sum_{j=1}^{|y_t|} r_t(\theta) \hat{A}_{t,j} \nabla_\theta \log \pi_\theta(y_{t,j} \mid x, m_{\leq t}, y_{<t}, y_{t,<j})$. Consider two continuations from the same prefix: a short trajectory $\tau^{\mathrm{S}}$ with horizon $T_{\mathrm{S}}$ and a long trajectory $\tau^{\mathrm{L}}$ with horizon $T_{\mathrm{L}} > T_{\mathrm{S}}$, leading to the same final reward. Define $\kappa \triangleq \frac{\|Z_t(\tau^{\mathrm{L}})\|}{\|Z_t(\tau^{\mathrm{S}})\|}$. If $\kappa < \frac{T_{\mathrm{L}}}{T_{\mathrm{S}}}$, then $\frac{\|g_t(\tau^{\mathrm{S}})\|}{\|g_t(\tau^{\mathrm{L}})\|} > 1$.*

This theorem shows that unless the aggregated contribution $Z_t(\tau^{\mathrm{L}})$ is at least $\frac{T_{\mathrm{L}}}{T_{\mathrm{S}}}$ times larger than $Z_t(\tau^{\mathrm{S}})$, the gradient update favors the trajectory with fewer turns. Importantly, this holds whether the advantages are both positive or both negative: in the latter case, although both trajectories are discouraged, the shorter one is penalized less. Consequently, the model is biased toward actions that reduce the number of turns, even if longer trajectories are equally rewarding. Empirically, our results in Appendix F show that reasoning processes with lazy-agent behavior (e.g., producing empty outputs or simply summarizing) consistently involve fewer turns than those without lazy agents at the initial training stages, which are critical in shaping policy behavior. *Together, our theorem and empirical findings explain the emergence of lazy-agent behavior*: the normalization bias steers optimization toward trajectories with fewer turns, and reasoning processes exhibiting lazy behavior naturally produce shorter trajectories, thereby receiving preferential reinforcement during training. We emphasize that our analysis is distinct from Dr.GRPO (Liu et al., 2025b), which examines token-level normalization. Their work removes length normalization to prevent the policy from favoring shorter correct answers or unnecessarily long incorrect ones. Furthermore, since the number of turns $T$ is far smaller than the number of tokens, the resulting bias in our setting is substantially more pronounced.

## 5.2 SHAPLEY-INSPIRED CAUSAL INFLUENCE

To mitigate the optimization bias in multi-turn GRPO, we first remove the $\frac{1}{T_i}$ normalization in Eq. 2, which alleviates but does not fully eliminate the lazy-agent issue as indicated in ablation study (Sec. 6.6). Addressing this problem requires measuring the causal influence of each step during online training. In practice, however, the policy generates only a single continuation per step, so influence must be inferred from one trajectory. This creates two challenges: **(1)** it offers only a limited view of how a step shapes the reasoning process (Xu et al., 2025), and **(2)** it biases causal influence toward specific phrasings rather than underlying ideas (Pavlick & Callison-Burch, 2016; McCoy et al., 2019; Merrick & Taly, 2020; Li et al., 2024). Analogous to Shapley values (Li et al., 2021), which attribute contributions by averaging marginal effects across all coalitions, step-level causal influence in multi-agent RL should reflect average contributions across possible continuations rather than a single path. Directly computing such Shapley-style values, however, is infeasible because it

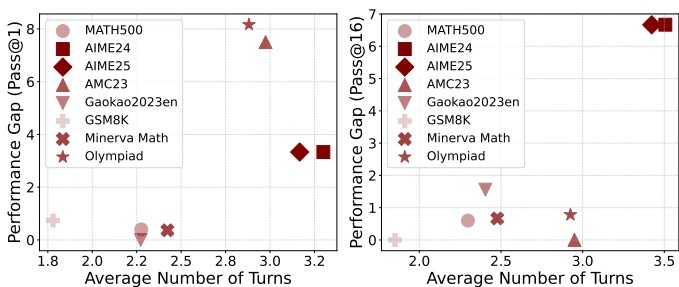

Figure 3: Performance gap between ReMA+ and ReMA across 8 benchmarks. Left: Pass@1. Right: Pass@16. Darker colors represent more difficult benchmarks.

would require extensive resampling during online RL. To make this tractable, we propose a stable Shapley-inspired causal influence measure.

We flatten each trajectory into a sequence $s_{i,1}, s_{i,2}, \ldots, s_{i,2T}$, where $s_{i,2t-1} = m_{i,t}$ (meta-thinking) and $s_{i,2t} = y_{i,t}$ (reasoning). Each step $s_{i,t}$ is treated as an *anchor step*, for which we form a group of semantically similar steps:

$$G_S(s_{i,t}) \;=\; \{\, s_{j,t'} \mid s_{j,t'} \approx s_{i,t},\; 1 \le j \le N,\; 1 \le t' \le 2T_j \,\},$$

where $\approx$ denotes semantic similarity. Similarity can be easily measured through semantic distance (see Appendix B), ensuring that steps within the same group express a comparable idea. For each step $s_{j,t'} \in G_S(s_{i,t})$, we measure its *one-step causal influence* on the next step. Let $h_{\le t'}$ be the local history up to and including step $t'$, and $h_{\le t'}^{(j)\backslash t'}$ the masked history with $s_{j,t'}$ removed. We compare the probability of the next output under the full and masked histories:

$$p_{\text{full}}^{(j,t')} \;=\; \pi_\theta(s_{j,t'+1} \mid h_{\le t'}^{(j)}), \quad p_{\text{mask}}^{(j,t')} \;=\; \pi_\theta(s_{j,t'+1} \mid h_{\le t'}^{(j)\backslash t'}), \quad \Delta\ell_{j,t'} \;\triangleq\; \log p_{\text{mask}}^{(j,t')} - \log p_{\text{full}}^{(j,t')}. \tag{4}$$

Finally, the Shapley-inspired causal influence of an anchor step $s_{i,t}$ is the average across its group:

$$\text{CI}(s_{i,t}) \;=\; \frac{1}{|G_S(s_{i,t})|} \sum_{(j,t')\,:\,s_{j,t'}\in G_S(s_{i,t})} \Delta\ell_{j,t'}. \tag{5}$$

Our method ensures reliable causal influence estimation by: **(1)** averaging the impact of semantically similar steps across rollouts to obtain a stable estimate of an idea's overall contribution, and **(2)** aggregating different phrasings of the same idea to reduce wording bias in influence estimates.

### 5.3 REASONING AGENT DELIBERATION FOR MULTI-TURN INTERACTIONS

As each agent contributes more actively, the number of dialogue turns between the meta-thinking and reasoning agents increases. However, prior work shows that longer multi-turn interactions can degrade performance: Laban et al. (2025) compare LLMs in (1) a single-turn setting where the full task is given in one prompt, and (2) a multi-turn setting where the task is decomposed into incremental prompts. They report consistent performance drops in the multi-turn condition, likely because LLMs overcommit to underspecified early context and struggle to recover from initial errors. These findings imply that while multi-agent collaboration can enrich reasoning, it also heightens vulnerability to error propagation when intermediate turns introduce ambiguity. If we view the meta-thinking agent as a user providing step-by-step instructions, then more interactions risk the reasoning agent becoming "lost" in dialogue, as observed by Laban et al. (2025). To mitigate this, we hypothesize that allowing the reasoning agent to discard prior responses, aggregate the meta-thinking prompts, and restart reasoning would be beneficial.

To validate this assumption, we conduct preliminary experiments following the ReMA framework to obtain a meta-thinking agent and a reasoning agent. We then compare the standard ReMA framework with a modified variant, **ReMA+**, where the reasoning agent is guided by a refined system prompt that enables it to adaptively discard its previous outputs when necessary. The full prompt design and additional details are provided in Appendix C.2. In this experiment, we modify the system

prompt only at inference time. Effectiveness is evaluated by (i) the performance gap in validation accuracy between ReMA+ and ReMA. Results across eight benchmarks are shown in Fig. 3. From the figure, we observe: **(1)** Without explicit training for deliberation, ReMA+ consistently matches or outperforms ReMA, with gains of about 8% on AMC23 and Olympiad under Pass@1, and 7% on AIME24 and AIME25 under Pass@16. **(2)** Even under Pass@16, which gives both frameworks ample chances to succeed, ReMA+ outperforms ReMA on 6 of 8 benchmarks—demonstrating that while LLMs have this capacity, explicit prompting is required for consistent behavior. **(3)** The performance gap widens as benchmark difficulty and the number of turns increase, highlighting the value of adaptive deliberation in extended multi-agent interactions.

Building on these observations, we design a method that trains the reasoning agent to adaptively discard its previous outputs to improve the likelihood of producing a correct final answer. Specifically, we introduce a control token, `<restart>`, which instructs the agent to discard prior responses, consolidate the instructions, and begin a fresh attempt. To assess the benefit of this mechanism, we develop a novel verifiable criterion that measures how discarding history affects the model's probability of generating the correct final output.

**Verifiable Reward for Restart.** Consider the $i$-th rollout where the reasoning agent outputs `<restart>` at turn $t$. In this case, we mask all reasoning-agent outputs that occur strictly before $t$: $\mathcal{Y}_{<2t}^{(i)} \triangleq \{s_{j,k} \mid k < 2t, \ k \bmod 2 = 0\}$, and define the causal influence of the restart action on the final reasoning step $\mathbf{y}_T^{(i)} = s_{i,2T}$ as

$$\Delta \ell_{i,t} \triangleq \log \pi_\theta(s_{i,2T} \mid h_{\leq 2T}^{(i)\backslash \mathcal{Y}_{<2t}^{(i)}}) \ - \ \log \pi_\theta(s_{i,2T} \mid h_{\leq 2T}^{(i)}). \tag{6}$$

We define a binary outcome reward $z_i$ as $+1$ if the final answer $\mathbf{y}_T^{(i)}$ is correct, and $-1$ if it is incorrect. The restart reward is then

$$r_{i,t}^{\text{restart}} = \begin{cases} +1, & \text{if } (z_i = +1 \ \wedge \ \Delta \ell_{i,t} > 0) \ \text{ or } \ (z_i = -1 \ \wedge \ \Delta \ell_{i,t} < 0), \\ -1, & \text{if } (z_i = +1 \ \wedge \ \Delta \ell_{i,t} < 0) \ \text{ or } \ (z_i = -1 \ \wedge \ \Delta \ell_{i,t} > 0), \\ 0, & \text{if } \Delta \ell_{i,t} = 0. \end{cases} \tag{7}$$

This reward provides a verifiable signal of whether the restart improves or worsens the model's belief in the final output. If the final answer is correct ($z_i = +1$), the restart is rewarded when masking prior reasoning increases confidence ($\Delta \ell_{i,t} > 0$); otherwise it is penalized. The converse holds when the final answer is incorrect ($z_i = -1$).

**Aggregated Step-Level Advantage.** Let $\text{CI}(s_{i,t})$ (Eq. 5) denote the Shapley-inspired causal influence for step $t$ in rollout $i$, and let $r_{i,t}^{\text{restart}}$ be the verifiable restart reward (Eq. 7), defaulting to 0 if no `<restart>` is issued. For normalization, each signal $x$ is first rescaled to $\widetilde{x} \in [-1, 1]$ using min–max scaling (details in Appendix B), and then standardized across all rollouts with mean $\mu_{\widetilde{X}}$ and standard deviation $\sigma_{\widetilde{X}}$ by $\mathcal{Z}_X(x) \triangleq \frac{\widetilde{x} - \mu_{\widetilde{X}}}{\sigma_{\widetilde{X}}}$. We apply this procedure to obtain the normalized causal signal $\tilde{C}_{i,t} = \mathcal{Z}_{\{\text{CI}\}}\big(\text{CI}(s_{i,t})\big)$ and restart signal $\tilde{R}_{i,t} = \mathcal{Z}_{\{r^{\text{restart}}\}}\big(r_{i,t}^{\text{restart}}\big)$. The overall step-level advantage is then defined as a weighted combination:

$$A_{i,t}^{\text{step}} = \tilde{A}_{i,t} + \alpha \tilde{C}_{i,t} + \beta \tilde{R}_{i,t}, \tag{8}$$

where $\tilde{A}_{i,t}$ is the normalized outcome-based advantage, and $\alpha, \beta$ are tunable hyperparameters. The training objective for Dr. MAMR builds on Eq. 2, removing the $\frac{1}{T}$ normalization and replacing the advantage function with Eq. 8. See implementation details in Appendix B.

# 6 EXPERIMENTS

## 6.1 EXPERIMENT SETTINGS

**Dataset and Benchmarks:** We conduct experiments on mathematical reasoning by training models on DeepScaleR dataset (Luo et al., 2025). The optimized agents are then evaluated across seven benchmarks: MATH500 (Lightman et al., 2023), GSM8K (Cobbe et al., 2021), AIME , AMC23

(Jia LI, 2024) , GaoKao2023En (Zhang et al., 2023a), Minerva Math (Lewkowycz et al., 2022), and OlympiadBench (He et al., 2024). See implementation details in Appendix B.

**Baselines and Models:** We compare Dr.MAMR against three baselines: (1) **GRPO** (Guo et al., 2025a), where the base model is trained with vanilla GRPO in a single-agent setting; (2) **VRP (CoT)** (Wan et al., 2025), where the base model is prompted step by step to operate within the meta-thinking and reasoning framework; and (3) **ReMA** (Wan et al., 2025), a multi-agent meta-reasoning framework trained with multi-turn GRPO. We conduct training and evaluation on the Qwen2.5 family, using the 3B, 7B, and 14B Instruct models (Team, 2024).

## 6.2 RESULTS ON SEVEN BENCHMARKS

**Question 1:** *How does Dr. MAMR perform in reasoning tasks compared to baselines?*

Table 1 reports pass@1 performance across seven benchmarks using 7B and 14B base models. (See Appendix H.6 for 3B results). From the results, we observe the following: **(1)** ReMA consistently underperforms compared to single-agent GRPO, highlighting the severity of the lazy agent issue and its detrimental effect on multi-agent performance. **(2)** Dr. MAMR consistently outperforms single-agent GRPO across all base models. Notably, the performance gain increases with larger base models that exhibit stronger instruction-following capabilities. This suggests that mitigating the lazy agent problem enables effective collaboration between agents and leads to better outcomes. This also points to a promising direction: improving the instruction-following ability of base models, which supports more effective communication and collaboration between agents, can in turn lead to better overall system performance. **(3) Our Dr. MAMR elevates the multi-agent system baseline from performing worse than single-agent GRPO to clearly outperforming it**, demonstrating the potential of multi-agent frameworks in solving complex reasoning tasks when carefully designed.

Table 1: Performance on math benchmarks.

| Model | Benchmark | GRPO | VRP (CoT) | ReMA | Dr. MAMR (Ours) |
|---|---|---|---|---|---|
| Qwen2.5 -7B -Instruct | MATH500 | 75.50 | 75.00 | 74.40 | **78.60** |
| | GSM8K | 90.50 | 92.04 | 90.60 | **92.12** |
| | AIME24 | 16.67 | 6.67 | 13.33 | **20.00** |
| | AMC23 | 55.00 | 47.50 | 50.00 | **62.50** |
| | Gaokao2023en | 64.60 | 56.62 | 57.92 | **65.20** |
| | Minerva Math | 34.70 | 35.66 | 34.93 | **38.24** |
| | Olympiad Bench | 48.60 | 38.22 | 42.58 | **52.34** |
| | **Average** | 55.08 | 50.24 | 51.97 | **58.43** |
| Qwen2.5 -14B -Instruct | MATH500 | **80.60** | 78.40 | 79.20 | 80.40 |
| | GSM8K | **94.50** | 92.87 | 93.59 | 93.69 |
| | AIME24 | 16.67 | 10.00 | 13.33 | **26.67** |
| | AMC23 | 60.00 | 55.00 | 60.00 | **67.50** |
| | Gaokao2023en | 64.90 | 66.23 | 67.53 | **69.09** |
| | Minerva Math | 41.50 | 38.60 | 41.91 | **43.02** |
| | Olympiad Bench | 48.20 | 46.78 | 45.12 | **57.03** |
| | **Average** | 58.05 | 55.41 | 57.24 | **62.49** |

## 6.3 TRAINING CURVES

**Question 2:** *How does the causal influence of agents evolve during training?*

In this section, we present a case study on the 7B model, examining how the causal influence of the meta-thinking agent and the reasoning agent evolves during training under our Dr. MAMR framework, compared to ReMA. We report the results in Fig. 4 (a). From the figure, we observe: **(1)** Under ReMA, the reasoning agent's causal influence initially increases slightly at the beginning of training but then steadily decreases, eventually approaching zero, while the meta-thinking agent's influence grows significantly as it comes to dominate the reasoning process. This indicates that naive

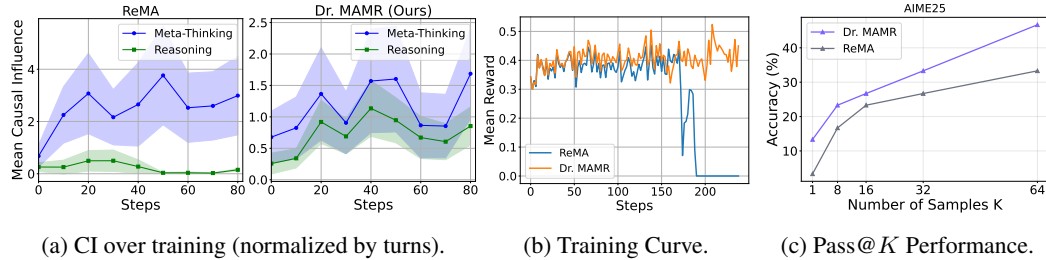

(a) CI over training (normalized by turns).    (b) Training Curve.    (c) Pass@$K$ Performance.

Figure 4: Results on causal influence, training stability, and pass@K.

multi-turn GRPO risks collapsing the system into an effective single-agent setup, losing the benefits of collaboration. **(2)** In contrast, under Dr. MAMR, the reasoning agent's influence steadily increases, while the meta-thinking agent also grow consistently, indicating that both agents contribute meaningfully. This balanced collaboration explains why Dr. MAMR achieves superior performance across diverse reasoning tasks compared to ReMA.

**Question 3:** *How does Dr. MAMR stabilize multi-agent RL compared to the baseline?*

Training stability is a key challenge in multi-agent RL. Thus, we compare the training curves and report mean rewards of Dr. MAMR and ReMA on the training data, with results for the 7B model shown in Fig. 4(b) and additional 3B results provided in Appendix H.7. From the figure, we observe that after 50 steps, Dr. MAMR achieves clearly superior performance. After 150 steps, ReMA collapses with its reward dropping to zero, whereas Dr. MAMR maintains stable training throughout. This demonstrates the benefit of addressing the lazy-agent issue for stabilizing multi-agent RL.

## 6.4 SCALING ON PASS@$K$

**Question 4.** *How does Dr. MAMR perform when scaling to pass@K?*

We examine test-time scaling (Muennighoff et al., 2025; Zhang et al., 2025a) by comparing the pass@$K$ performance of Dr. MAMR and ReMA, which measures whether the correct solution appears within the best result of $K$ independent attempts. Results on the most challenging benchmark, AIME25, are presented in Fig. 4(c), with additional benchmarks reported in Appendix H.9. The figure shows that **the performance gap between Dr. MAMR and ReMA widens as $K$ increases, highlighting Dr. MAMR's strength in tackling difficult tasks**.

## 6.5 TRAINING DYNAMICS OF RESTART BEHAVIOR

In this section, we provide a detailed analysis of the training dynamics associated with the restart behavior in Dr. MAMR. Following the experimental setting described in Sec. 6.1, we set $\alpha = \beta = 0.1$, use 8 rollouts per prompt, and adopt a batch size of 128. At each global step, we track three quantities: (1) the mean reward obtained on the training batch, (2) the total number of restart actions triggered by the reasoning agent, and (3) the average reward assigned to those restart actions. The results are visualized in Fig. 5. From the figure, several important patterns emerge.

First, as training progresses, the overall mean reward of Dr. MAMR increases steadily, and the mean reward associated specifically with restart actions exhibits a similar upward trend. This indicates that the restart reward signal is informative: it provides consistent and meaningful feedback to the reasoning agent, rather than introducing noise or instability. The agent learns to exploit the restart mechanism in a way that aligns with improved task performance. Second, although the number of restart actions experiences a mild increase during the earliest phase of training—an expected effect due to the restart reward signal not yet being fully calibrated—this trend reverses quickly. After approximately 40 training steps, the number of restart actions begins to decline and eventually stabilizes around step 100. While the restart frequency does decrease, the reduction is only partial, indicating that the agent is learning to avoid unnecessary or redundant restarts. Importantly, this stabilization occurs despite the fact that the reward for restart actions continues to increase. In other words, the positive restart reward does not cause the agent to enter a degenerate mode where restarts

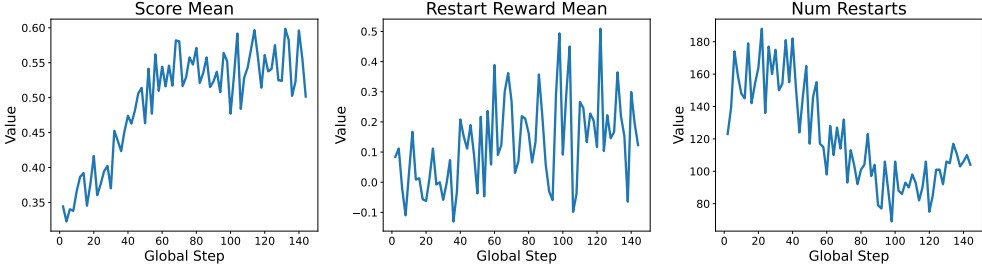

Figure 5: Training dynamics of restart behavior.

are over-triggered simply to accumulate reward. This observation is precisely the behavior we aim to achieve: the reasoning agent is not "hacked" into producing restart tokens blindly. Instead, it learns when a restart is genuinely beneficial based on the current prompt and the intermediate reasoning trajectory.

### 6.6 ABLATION STUDY

**Question 5.** *How does each component contribute to reasoning performance?*

We compare the full model against three variants: (1) w/o Normalization Debias (w/o ND), which retains the turn-level normalization from ReMA; (2) w/o Shapley-inspired causal influence (w/o CI); and (3) w/o Restart Behavior (w/o RB). We train these variants and report their benchmark performance in Table 6. From the table, we observe: **(1)** Dropping either normalization or causal influence causes clear performance drops, showing their complementary role in discouraging shortcuts and promoting balanced contributions. **(2)** Removing restart behavior also degrades performance across benchmarks, though less severely, highlighting its value in enabling recovery from mistakes and sustaining stable reasoning. We provide a case study on restart behavior in Appendix H.8.

Table 2: Ablation study on the 7B model.

| Variant | AIME24 | AMC23 | Gaokao2023en | Olympiad Bench |
|---------|--------|-------|--------------|----------------|
| Dr.MAMR | 20.00 | 62.50 | 65.20 | 52.34 |
| w/o ND | 13.33 | 55.00 | 63.64 | 47.85 |
| w/o CI | 13.33 | 52.50 | 63.38 | 45.31 |
| w/o RB | 16.67 | 57.50 | 63.90 | 50.58 |

## 7 CONCLUSION

We identify the issue of lazy agents in multi-agent LLM reasoning and trace the issue to the loss structure of multi-turn GRPO. To address it, we introduce a Shapley-inspired causal influence measure and a verifiable reward for restart behavior. Experiments across diverse benchmarks demonstrate that Dr. MAMR effectively mitigates lazy behavior and surpasses strong single- and multi-agent baselines, unlocking the potential of multi-agent frameworks for complex reasoning tasks.

### ACKNOWLEDGMENT

This material is based upon work supported by, or in part by the Army Research Office (ARO) under grant number W911NF-21-10198 and the Department of Homeland Security (DHS) under grant number 17STCIN00001-05-00. The views and conclusions contained in this material are those of the authors and should not be interpreted as necessarily representing the official policies, either expressed or implied, of the funding agencies.

ETHICS STATEMENT

This work does not involve human subjects or the collection of new datasets. Experiments use established corpora and benchmarks under their licenses.

REPRODUCIBILITY STATEMENT

We provided a detailed description implementation details in Appendix B and experimental settings in Appendix C.

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

# A DETAILED RELATED WORKS

## A.1 MULTI-AGENT RL

Multi-agent RL addresses how multiple agents coordinate in a shared environment to maximize collective performance, with a central challenge being credit assignment: determining each agent's contribution to the overall reward. Classical solutions include value decomposition (VDN; Sunehag et al., 2018), counterfactual baselines (COMA; Foerster et al., 2018), regression-based reward functions with default-action substitution (Dr.Reinforce; Castellini et al., 2022), social influence estimation via KL divergence (Jaques et al., 2019), role-based coordination through role networks (Wang et al., 2020), and model-based transition prediction to measure influence (Liu et al., 2023). With the advent of large language model (LLM) agents, MARL techniques have been extended to multi-turn reasoning and cooperative dialogue: turn-level credit assignment to reduce misattributed rewards (Zeng et al., 2025), SWEET-RL for critic-driven step-wise rewards (Zhou et al., 2025), MARFT for alleviating agent inactivity and communication inefficiency (Liao et al., 2025), RAGEN for addressing the "Echo Trap" caused by coarse reward signals (Wang et al., 2025), and MAGRPO for framing LLM collaboration as cooperative MARL with tailored reward design (Liu et al., 2025a). A persistent challenge across both classical and LLM-based MARL is the emergence of lazy agents that contribute little while relying on others. Recent work therefore explores causal influence estimation (Bogdan et al., 2025; Nguyen et al., 2025; Liu et al., 2024b), introducing methods such as black-box resampling and attention suppression to quantify how an agent's utterance shapes subsequent decisions, thereby enabling finer-grained credit assignment and mitigating agent inactivity.

## A.2 HIERARCHICAL RL

Hierarchical multi-agent systems coordinate cooperation by assigning high-level controllers to decompose tasks for lower-level workers, a design shown to improve scalability, robustness, and long-horizon reasoning. (Yu et al., 2024) introduce a manager–analyst paradigm for structured decomposition, while Chain-of-Agents (CoA) leverages chained communication to approximate hierarchical coordination with greater flexibility (Zhang et al., 2024). Empirical studies further demonstrate that boss–worker hierarchies outperform flat or linear structures under failure conditions (Huang et al., 2024). Memory-oriented approaches, exemplified by Tracing Hierarchical Memory for Multi-Agent Systems, highlight how layered storage and retrieval mechanisms enable adaptive long-horizon collaboration (Zhang et al., 2025b). On the planning side, Hierarchical Planning for Complex Tasks with Knowledge-Graph RAG and Symbolic Verification integrates structured decomposition with formal verification (Cornelio et al., 2025). More explicit architectures, such as HMAW and ReAc-Tree, explore CEO–Manager–Worker hierarchies or adaptive tree structures for general task allocation (Liu et al., 2024c; Choi et al., 2025). Hierarchical reinforcement learning (HRL) organizes control as high-level planning over subgoals with low-level execution, enabling agents to operate over long horizons via temporal abstraction and reusable skills. Early neural HRL emphasized top-down goal setting—e.g., high-level goal embeddings steering a low-level policy (Vezhnevets et al., 2017) and quickly expanded toward unsupervised skill discovery to populate the low-level option set (Bagaria & Konidaris, 2019). Beyond purely reactive control, symbolic reasoning has been fused with HRL to support plan construction and revision (Yamamoto et al., 2018). Recent advances improve the reliability and interpretability of options themselves: programmatic, human-readable sub-policies selected by the high-level planner enhance generalization to longer tasks (Lin et al., 2024), while theoretically grounded supervision clarifies how limited human feedback can be efficiently allocated across hierarchical levels in goal-conditioned settings (Yan & Lipton, 2024). Complementary lines strengthen the low level: disentangling unsupervised skill discovery yields cleaner building blocks (Hu et al., 2024), and integrating imperfect expert priors improves multi-agent coordination under hierarchical control (Liu et al., 2024a).

## A.3 LLM REASONING

Large Language Models (LLMs) have demonstrated strong performance across a wide range of natural language tasks (Brown et al., 2020; Chowdhery et al., 2023; Du et al., 2022; Dubey et al., 2024; Wenzek et al., 2019). Early research found that prompting models to reason step by step, an approach known as chain-of-thought (CoT) prompting, can significantly improve performance on arithmetic, commonsense, and symbolic reasoning tasks by eliciting intermediate reasoning steps (Wei et al.,

2022; Kojima et al., 2022; Nye et al., 2021). Building on this, researchers have explored non-linear reasoning structures. For example, Tree-of-Thoughts (ToT) organizes candidate reasoning paths into a search tree with lookahead capabilities, while Graph-of-Thoughts (GoT) generalizes reasoning to arbitrary graphs of "thought" nodes and edges, expanding the space for structured deliberation (Yao et al., 2023; Besta et al., 2024). These advances have inspired the development of Large Reasoning Models (LRMs), models explicitly trained for multi-step reasoning (Guo et al., 2025a; Achiam et al., 2023; Grattafiori et al., 2024; Xu et al., 2023; Zhou et al., 2022; Wu et al., 2024b; Qi et al., 2024; Chae et al., 2024). Typically, LRMs undergo supervised fine-tuning followed by a reinforcement learning stage, and have achieved state-of-the-art results on challenging tasks such as math, coding, and task planning (Jaech et al., 2024; Guo et al., 2025a; Comanici et al., 2025; Yang et al., 2024a; 2025; Lightman et al., 2023; Wang et al., 2023). The success of strong single-model reasoners has also spurred multi-agent approaches, where complex tasks are decomposed and coordinated among specialized LLM agents via role assignment, orchestration, and debate, mirroring human teamwork (Li et al., 2023; Wu et al., 2024a; Chen et al., 2023; Du et al., 2023; Yuan & Xie).

# B  IMPLEMENTATION DETAILS

## B.1  TRAINING OBJECTIVE OF DR. MAMR

The training objective for Dr. MAMR builds on Eq. 2 with two key modifications: (i) we remove the $\frac{1}{T_i}$ normalization over the number of turns, and (ii) we replace the step-level advantage with the weighted formulation in Eq. 8. The resulting objective is defined as follows:

$$
\mathcal{J}_{\text{Dr. MAMR}}(\theta) = \mathbb{E}_{(\mathbf{x},\mathbf{y}^*)\sim\mathcal{D},\,\{(\mathbf{m}_i,\mathbf{y}_i)\}_{i=1}^{G}\sim\pi_{\theta_{\text{old}}}(\cdot|\mathbf{x})}
$$
$$
\left[ \frac{1}{G} \sum_{i=1}^{G} \sum_{t=1}^{T_i} \frac{1}{|\mathbf{y}_{i,t}|} \sum_{j=1}^{|\mathbf{y}_{i,t}|} \Big( \min \big( r_{i,t}(\theta) A_{i,t}^{\text{step}}, \text{clip}(r_{i,t}(\theta), 1-\epsilon, 1+\epsilon) A_{i,t}^{\text{step}} \big) - \beta\, D_{\text{KL}}(\pi_\theta \| \pi_{\text{ref}}) \Big) \right],
$$
$$
(9)
$$

where the step-level advantage is given by Eq. 8:

$$
A_{i,t}^{\text{step}} = \tilde{A}_{i,t} + \alpha \tilde{C}_{i,t} + \beta \tilde{R}_{i,t}.
$$

## B.2  TRAINING ON DEEPSCALER

We conduct all experiments using the Verl RL framework (Sheng et al., 2024). Given the substantial computational cost, we fix the hyperparameters at $\alpha = \beta = 0.1$ across all experiments, as this setting provides stable performance. We use bfloat16 precision for training, with a batch size of 128 and 128 sampled rollouts per training step.

## B.3  SHAPLEY-INSPIRED CAUSAL INFLUENCE

To group semantically similar steps for causal influence estimation, we use Qwen2.5-0.5B [1] as the embedding model. Each step $s_{i,t}$ in a trajectory is encoded into a dense vector representation, and semantic similarity between steps is measured using cosine similarity. For each anchor step, we form a group $G_S(s_{i,t})$ by including all steps whose embeddings have cosine similarity of at least 0.9 with the anchor. This threshold ensures that only highly similar steps—those expressing essentially the same idea, regardless of minor wording differences—are grouped together.

## B.4  COLD START FOR META-THINK AND REASONING

For the experiments in Table 1, we adopt RL-from-base since the Qwen2.5B-Instruct family (Team, 2024) demonstrates strong instruction-following capability. Moreover, as shown in ReMA (Wan et al., 2025), the performance gap between RL-from-base and RL-from-SFT is marginal.

---

[1]`https://huggingface.co/Qwen/Qwen2.5-0.5B`

## B.5 Cold start for restart behavior

However, it is difficult for the base model to exhibit restart behavior directly. To address this, we collect expert data and apply supervised fine-tuning (SFT) to the base model. Specifically, building on the meta-thinking and reasoning datasets collected by ReMA (Wan et al., 2025; Ye et al., 2025), we use GPT-4o (Hurst et al., 2024) to adversarially insert a few steps of noisy reasoning, followed by a restart, in order to simulate scenarios where the reasoning agent becomes lost in the conversation and the restart behavior enables recovery. For supervised fine-tuning, we use the LlamaFactory codebase, training the model for 3 epochs with a learning rate of 1e-5, a cosine learning rate scheduler, and a batch size of 8. We employ DeepSpeed ZeRO-2 for distributed training.

## C More Experimental Details

### C.1 Preliminary experiments on causal influence

#### C.1.1 Experimental setup

We follow the setting in (Wan et al., 2025) and train models on 7.5k training samples in MATH (Hendrycks et al., 2021) and test on datasets: GSM8K (Cobbe et al., 2021), AIME24[2], AMC23[3], GaoKao2023En (Zhang et al., 2023a), Minerva Math (Lewkowycz et al., 2022), and Olympiad Bench (He et al., 2024). We generate reasoning processes on the evaluation benchmark and subsequently measure the causal influence of each response following the descriptin in Sec. 4.

#### C.1.2 Prompts

---

**System Prompts for Meta-Think and Reasoning Agents of ReMA**

**META-THINK AGENT SYSTEM PROMPT**
You are a meta-think agent that represents human high-level thinking processes. When solving a question, you will have a discussion with a human. Each time, think about what to do next. For example:

- Exploring multiple angles and approaches
- Breaking down the solution into clear steps
- Continuously reflecting on intermediate results honestly and adapting your strategy as you progress
- Backtracking when necessary
- Requesting exploration of multiple solutions individually
- Finally, confirm the answer with the tag [FINISH]

**REASONING AGENT SYSTEM PROMPT**
Please reason step by step following the given instruction. When asked to finalize your answer, put your answer within \boxed{}.

---

[2]https://huggingface.co/datasets/AI-MO/aimo-validation-aime
[3]https://huggingface.co/datasets/AI-MO/aimo-validation-amc

---

**Refined System Prompts for Meta-Think and Reasoning Agents**

**META-THINK AGENT SYSTEM PROMPT**
You are a meta-think agent that represents human high-level think process. When solving a question, you will have a discussion with the human, and each time you will think about what to do next. For example:

- Exploring multiple angles and approaches
- Breaking down the solution into clear steps
- Continuously reflecting on intermediate results honestly and adapting your strategy as you progress
- Backtracking when necessary
- Requesting exploration of multiple solutions individually
- Finally, confirm the answer with the tag `[FINISH]`.

**Please *do not* focus on completing the task by calculating the final answer**; that step will be handled by a separate reasoning agent.

**REASONING AGENT SYSTEM PROMPT**
You are a reasoning agent that follows structured problem-solving instructions step by step. Your goals are:

- Follow the given instruction precisely.
- Reason step by step toward a solution.
- Avoid producing empty or blank outputs at any step.
- If uncertain, provide your best reasoning and partial answer rather than outputting nothing.
- Always provide a meaningful and non-empty response, even during intermediate steps.
- When you receive the signal `[FINISH]`, finalize your answer and place it within `\boxed{}`.
- If unable to finalize, explain why and still output your best available answer within `\boxed{}`.

**Remember: You must never produce trivial outputs.**

---

## C.2   PRELIMINARY EXPERIMENTS ON HOW DELIBERATION HELPS MULTI-TURN REASONING

In this section, we provide additional details about the preliminary experiments referenced in Sec. 5.3, which demonstrate that deliberation can enhance performance in multi-turn reasoning tasks.

We adopt the experimental setup outlined in Sec. C.1.1 to train both the meta-thinking agent and the reasoning agent using the original ReMA framework (Wan et al., 2025). To evaluate the impact of deliberation, we introduce a variant of ReMA at inference time by modifying the system prompt of the reasoning agent as follows:

---

**System Prompt for Reasoning Agent of ReMA+**

Please reason step by step. When asked to finalize your answer, place your result within `\boxed{}`.

**Deliberation Instruction**
If your previous reasoning is based on assumptions that are contradicted or made incomplete by new instructions, you must explicitly output:
`<restart>`
Reason: `<One-sentence justification>`

**Rethinking Rules for New Information**

1. Upon receiving new information, re-evaluate all prior assumptions, steps, and conclusions.

2. If the new information invalidates or alters your earlier reasoning, clearly acknowledge this and issue a `<restart>`.

3. Do not simply add new information to prior reasoning — verify that the existing logic still holds.

4. Ask clarifying questions only if essential information is missing and further reasoning is blocked.

5. If you revise your reasoning due to new information, explicitly state:
   `"Given this new information, I need to revise my approach."`

---

## D  CASE STUDY ON LAZY BEHAVIOR

---

**Case Study**

`[Question]`
Determine the number of ways to arrange the letters of the word 'ELLIPSE'. **[Groundtruth: 1260]**

**ReMA**

**Meta-thinking:**
Let's start by considering the total number of letters and any repetitions. The word ELLIPSE has 7 letters, with L repeated twice. To find the number of distinct arrangements, we use the formula for permutations of a multiset:

$$\frac{n!}{n_1! \cdot n_2! \cdot \ldots \cdot n_k!}$$

For ELLIPSE: - $n = 7$ - L appears 2 times - E, I, P, S each appear once

$$\frac{7!}{2!} = \frac{5040}{2} = 2520$$

**Reasoning:**

☐

**Meta-thinking:**

[FINISH]

**Reasoning:**

2520

---

> **Dr. MAMR**
>
> **Meta-thinking:**
> To determine the number of ways to arrange the letters of the word "ELLIPSE", we need to consider the total number of letters and any repetitions of letters.
> 1. First, let's count the total number of letters and identify any repeated letters.
> 2. Then, we'll use the formula for permutations of a multiset.
> Let's start by counting the letters. How many letters are there in the word "ELLIPSE"? And which letters are repeated?
>
> **Reasoning:**
> The word "ELLIPSE" consists of 7 letters: E, L, L, I, P, S, E. We can see that the letter 'E' is repeated twice, and the letter 'L' is also repeated twice. The other letters, I, P, and S, each appear once. To determine the number of distinct arrangements of these letters, we use the formula for permutations of a multiset:
>
> $$\frac{n!}{n_1! \cdot n_2! \cdot \ldots \cdot n_k!}$$
>
> For the word "ELLIPSE": - $n = 7$ - 'E' occurs 2 times, 'L' occurs 2 times, others occur once
>
> $$\frac{7!}{2! \cdot 2! \cdot 1! \cdot 1! \cdot 1!} = \frac{5040}{4} = 1260$$
>
> $$\boxed{1260}$$
>
> **Meta-thinking:**
> Great job walking through the calculation! Let's summarize:
>
> - 7 letters in total
>
> - 'E' and 'L' both appear twice
>
> - Used multiset formula correctly
>
> - Answer: $\boxed{1260}$

## E  PROOFS

Following Eq.2, ReMA's training objective (per batch) is:

$$\mathcal{J}(\theta) = \mathbb{E}\left[\frac{1}{G}\sum_{i=1}^{G}\frac{1}{T_i}\sum_{t=1}^{T_i}\frac{1}{|y_{i,t}|}\sum_{j=1}^{|y_{i,t}|}\left(\min\{u_{i,t,j}, v_{i,t,j}\} - \beta\, D_{\mathrm{KL}}(\pi_\theta \,\|\, \pi_{\mathrm{ref}})\right)\right], \qquad (10)$$

where

$$u_{i,t,j} = r_{i,t}(\theta)\,\hat{A}_{i,t,j}, \qquad (11)$$
$$v_{i,t,j} = \mathrm{clip}(r_{i,t}(\theta), 1-\epsilon, 1+\epsilon)\,\hat{A}_{i,t,j}. \qquad (12)$$

Turn-level importance ratio:

$$r_{i,t}(\theta) = \frac{1}{|y_{i,t}|}\sum_{j'=1}^{|y_{i,t}|}\underbrace{\frac{\pi_\theta(y_{i,t,j'} \mid c_{i,t,j'})}{\pi_{\theta_{\mathrm{old}}}(y_{i,t,j'} \mid c_{i,t,j'})}}_{=:\, r_{i,t,j'}(\theta)}, \qquad (13)$$

where the context is:

$$c_{i,t,j'} := (x_i, \{m_{i,\cdot}, y_{i,\cdot}\}_{<t}, m_{i,t}, y_{i,t,<j'}).$$

We derive the turn-$t$ contribution for a fixed trajectory $i$, i.e.,

$$L_{i,t}(\theta) := \frac{1}{T_i}\cdot\frac{1}{|y_{i,t}|}\sum_{j=1}^{|y_{i,t}|}\min\{u_{i,t,j}, v_{i,t,j}\} - \frac{\beta}{T_i}\cdot\frac{1}{|y_{i,t}|}\sum_{j=1}^{|y_{i,t}|}D_{\mathrm{KL}}\left(\pi_\theta(\cdot \mid c_{i,t,j}) \,\|\, \pi_{\mathrm{ref}}(\cdot \mid c_{i,t,j})\right). \qquad (14)$$

Without considering the KL divergence and clipping,

$$L_{i,t}(\theta) = \frac{1}{T_i} \cdot \frac{1}{|y_{i,t}|} \sum_j r_{i,t}(\theta) \, \hat{A}_{i,t,j} = \frac{1}{T_i} \cdot \bar{A}_{i,t} \cdot r_{i,t}(\theta), \tag{15}$$

where

$$\bar{A}_{i,t} := \frac{1}{|y_{i,t}|} \sum_{j=1}^{|y_{i,t}|} \hat{A}_{i,t,j}. \tag{16}$$

Compute the gradient:

$$\begin{aligned}
\nabla_\theta r_{i,t}(\theta) &= \nabla_\theta \left( \frac{1}{|y_{i,t}|} \sum_{j'=1}^{|y_{i,t}|} r_{i,t,j'}(\theta) \right) \\
&= \frac{1}{|y_{i,t}|} \sum_{j'=1}^{|y_{i,t}|} \nabla_\theta \left( \frac{\pi_\theta(y_{i,t,j'} \mid c_{i,t,j'})}{\pi_{\theta_{\text{old}}}(y_{i,t,j'} \mid c_{i,t,j'})} \right) \\
&= \frac{1}{|y_{i,t}|} \sum_{j'=1}^{|y_{i,t}|} \frac{1}{\pi_{\theta_{\text{old}}}(y_{i,t,j'} \mid c_{i,t,j'})} \nabla_\theta \pi_\theta(y_{i,t,j'} \mid c_{i,t,j'}) \\
&= \frac{1}{|y_{i,t}|} \sum_{j'=1}^{|y_{i,t}|} \frac{\pi_\theta(y_{i,t,j'} \mid c_{i,t,j'})}{\pi_{\theta_{\text{old}}}(y_{i,t,j'} \mid c_{i,t,j'})} \nabla_\theta \log \pi_\theta(y_{i,t,j'} \mid c_{i,t,j'}) \\
&= \frac{1}{|y_{i,t}|} \sum_{j'=1}^{|y_{i,t}|} r_{i,t,j'}(\theta) \, \nabla_\theta \log \pi_\theta(y_{i,t,j'} \mid c_{i,t,j'}).
\end{aligned} \tag{17}$$

Therefore, the exact per-turn gradient (no clipping, no KL) is:

$$\nabla_\theta L_{i,t}(\theta) = \frac{1}{T_i} \cdot \bar{A}_{i,t} \cdot \left( \frac{1}{|y_{i,t}|} \sum_{j'=1}^{|y_{i,t}|} r_{i,t,j'}(\theta) \, \nabla_\theta \log \pi_\theta(y_{i,t,j'} \mid c_{i,t,j'}) \right) \tag{18}$$

We define the aggregated turn-$t$ stochastic contribution as

$$Z_t(\tau) \triangleq \frac{1}{|y_t|} \sum_{j=1}^{|y_t|} r_t(\theta) \, \hat{A}_{t,j} \, \nabla_\theta \log \pi_\theta(y_{t,j} \mid x, m_{\leq t}, y_{<t}, y_{t,<j}). \tag{19}$$

Then the ReMA gradient contribution at turn $t$ is

$$g_t(\tau) = \frac{1}{T(\tau)} Z_t(\tau). \tag{20}$$

Let

$$\kappa \triangleq \frac{\|Z_t(\tau^{\text{L}})\|}{\|Z_t(\tau^{\text{S}})\|}.$$

Then the relative gradient magnitude satisfies

$$\frac{\|g_t(\tau^{\text{S}})\|}{\|g_t(\tau^{\text{L}})\|} = \frac{T_{\text{L}}}{T_{\text{S}}} \cdot \frac{1}{\kappa}.$$

In particular, if $\kappa < \frac{T_{\text{L}}}{T_{\text{S}}}$, then

$$\frac{\|g_t(\tau^{\text{S}})\|}{\|g_t(\tau^{\text{L}})\|} > 1.$$

## F   PRELIMINARY EXPERIMENTS FOR THEORETICAL ANALYSIS

We present preliminary results on the mean number of turns in the reasoning process for cases with empty outputs and trivially copy the other's response (i.e., reasoning process exhibiting lazy-agent behavior) and those without empty outputs (i.e., reasoning process without lazy agents). We report results from the first 20 training steps, as this initial stage is critical in shaping the agent's behavior. As shown in Fig. 6, the number of turns for reasoning process exhibiting lazy-agent behavior is consistently smaller than that of non-lazy agents.

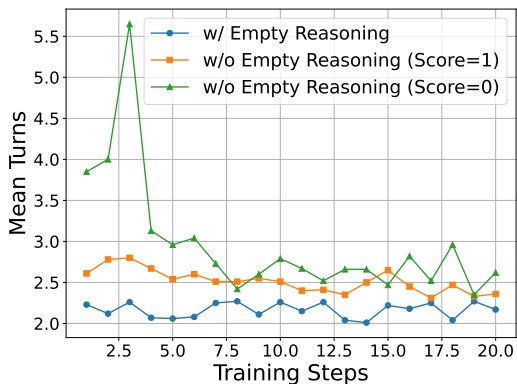

Figure 6: Mean number of turns comparing reasoning processes w/ and w/o lazy-agent behavior.

## G   PROCESS REWARD FAILS TO MITIGATE THE LAZY AGENT ISSUE

In this section, we present experimental results showing that using a process reward model to assign credit at each turn fails to mitigate the lazy agent issue in ReMA. We follow the experimental setup described in Section C.1, and adopt `Qwen/Qwen2.5-Math-PRM-7B`[4] as the process reward model (PRM). We revise Eq. 3 as follows:

$$r_{i,t}(\theta) = r_{i,t}(\theta) + a_{i,t}, \tag{21}$$

where $a_{i,t}$ denotes the process reward for the $t$-th turn in the $i$-th rollout, provided by the PRM.

The training curve of this approach is shown in Fig. 7. As observed, the model collapses rapidly after only 30 training steps. We attribute this to reward hacking, a well-known failure mode in RLHF settings, as documented in prior work (Zhang et al., 2025d; Gao et al., 2023; Yang et al., 2024b; Li et al., 2025a).

## H   ADDITIONAL EXPERIMENT RESULTS

### H.1   RESULTS ON ADDITIONAL MODEL FAMILY

In this section, following the experimental setup described in Sec. 6.1, we conduct experiments using two base models: Meta-Llama-3-8B-Instruct[5] and Llama-3.1-8B-Instruct[6] . We compare our proposed Dr. MAMR framework against ReMA and the single-agent GRPO baseline. The results are summarized in Table 3.

Across both Llama base models—which overall exhibit lower raw performance than the Qwen2.5 series—we consistently observe that Dr. MAMR yields substantial improvements over both ReMA and GRPO. This demonstrates that the benefits of our multi-agent meta-reasoning framework persist even with weaker base models, highlighting its robustness and general applicability.

---

[4]Qwen/Qwen2.5-Math-PRM-7B

[5]meta-llama/Meta-Llama-3-8B-Instruct

[6]meta-llama/Meta-Llama-3.1-8B-Instruct

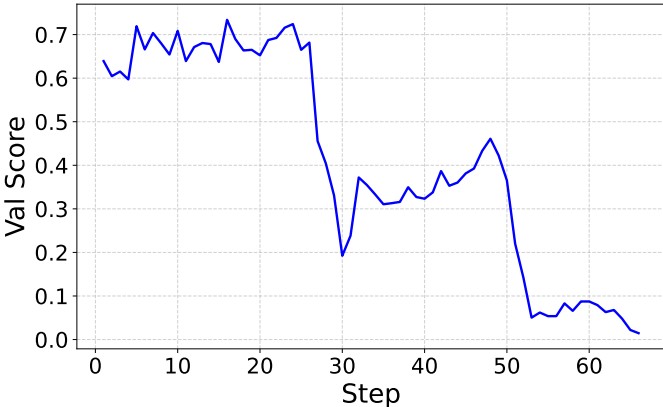

Figure 7: Training curve of ReMA with process reward assigned for each turn.

Table 3: Performance on math reasoning benchmarks (Llama family).

| Model | Benchmark | GRPO | ReMA | Dr. MAMR |
|-------|-----------|------|------|----------|
| **Llama3-8B-Instruct** | MATH500 | 40.4 | 33.80 | 43.20 |
| | GSM8K | 82.79 | 79.38 | 83.47 |
| | AIME24 | 0.00 | 0.00 | 3.33 |
| | AMC23 | 25.00 | 22.50 | 32.50 |
| | Gaokao2023en | 29.87 | 28.57 | 32.49 |
| | Minerva Math | 17.28 | 13.97 | 22.06 |
| | Olympiad Bench | 10.52 | 8.89 | 16.30 |
| | **Average** | **29.4** | **26.73** | **33.34** |
| **Llama3.1-8B-Instruct** | MATH500 | 55.20 | 53.20 | 58.60 |
| | GSM8K | 88.32 | 87.26 | 90.22 |
| | AIME24 | 10.00 | 6.67 | 13.33 |
| | AMC23 | 27.50 | 20.00 | 30.50 |
| | Gaokao2023en | 44.94 | 37.14 | 48.95 |
| | Minerva Math | 32.35 | 28.31 | 36.77 |
| | Olympiad Bench | 16.44 | 19.56 | 22.67 |
| | **Average** | **39.25** | **36.97** | **43.01** |

## H.2 HYPERPARAMETER ANALYSIS

In this section, we analyze the sensitivity of Dr. MAMR to its two hyperparameters: the weight for the causal-influence reward ($\alpha$) and the weight for the restart-behavior reward ($\beta$). Following the experimental setup in Sec. 6.1, we conduct two sets of experiments: (1) Fix $\alpha = 0.1$ and vary $\beta \in \{0.1, 0.3, 0.5\}$. (2) Fix $\beta = 0.1$ and vary $\alpha \in \{0.1, 0.3, 0.5\}$. The results are shown in Tables 4 and 5.

**(1) Effect of varying $\beta$ (restart-behavior reward).** When $\beta$ increases from 0.1 to 0.3, Dr. MAMR maintains very stable average performance (58.43 → 58.16). Even when $\beta$ is further increased to 0.5, where performance drops to 53.15, Dr. MAMR still outperforms ReMA (51.97) by a clear margin.

*Interpretation.* $\beta$ affects only the reasoning agent's restart behavior. A moderate value of $\beta$ (0.1 to 0.3) guides this behavior without disrupting the main optimization process. However, an excessively large $\beta$ risks over-encouraging or over-penalizing restarts, which discourages beneficial exploratory steps and ultimately leads to degraded performance.

**(2) Effect of varying $\alpha$ (causal-influence reward).** Increasing $\alpha$ from 0.1 to 0.3 yields only a mild change in performance (58.43 to 56.30). Even at $\alpha = 0.5$, where the score is 52.77, Dr. MAMR

maintains performance above the ReMA baseline (51.97), indicating that the method remains robust across a broad range of $\alpha$ values.

*Interpretation.* $\alpha$ encourages each agent to take actions that can influence the other's internal reasoning trajectory. When $\alpha$ becomes large, this incentive may occasionally place more emphasis on cross-agent influence than on improving final reasoning quality. As a result, the overall performance tends to vary more noticeably with respect to $\alpha$ than to $\beta$.

**(3) Overall robustness.** Across all combinations of $\alpha$ and $\beta$, mild hyperparameter choices (e.g., $\alpha = 0.1$ with $\beta = 0.3$, or $\beta = 0.1$ with $\alpha = 0.3$) achieve performance comparable to the best setting of $\alpha = 0.1, \beta = 0.1$. Even when the hyperparameters are set to larger values such as 0.5, Dr. MAMR still surpasses ReMA, as Dr. MAMR addresses the fundamental lazy-agent issue. Overall, these results show that Dr. MAMR provides robust improvements without requiring extensive hyperparameter tuning.

Table 4: Hyperparameter analysis with $\alpha = 0.1$.

| Model | Benchmark | ReMA | Dr. MAMR ($\beta = 0.1$) | ($\beta = 0.3$) | ($\beta = 0.5$) |
|---|---|---|---|---|---|
| Qwen2.5 -7B -Instruct | MATH500 | 74.40 | 78.60 | 78.80 | 77.20 |
| | GSM8K | 90.60 | 92.12 | 91.59 | 86.05 |
| | AIME24 | 13.33 | 20.00 | 20.00 | 16.67 |
| | AMC23 | 50.00 | 62.50 | 60.00 | 52.50 |
| | Gaokao2023en | 57.92 | 65.20 | 66.50 | 63.12 |
| | Minerva Math | 34.93 | 38.24 | 38.60 | 34.56 |
| | Olympiad Bench | 42.58 | 52.34 | 51.60 | 45.04 |
| | Average | 51.97 | 58.43 | 58.16 | 53.15 |

Table 5: Hyperparameter analysis with $\beta = 0.1$.

| Model | Benchmark | ReMA | Dr. MAMR ($\alpha = 0.1$) | ($\alpha = 0.3$) | ($\alpha = 0.5$) |
|---|---|---|---|---|---|
| Qwen2.5 -7B -Instruct | MATH500 | 74.40 | 78.60 | 77.80 | 76.80 |
| | GSM8K | 90.60 | 92.12 | 91.18 | 88.10 |
| | AIME24 | 13.33 | 20.00 | 16.67 | 13.33 |
| | AMC23 | 50.00 | 62.50 | 57.50 | 52.50 |
| | Gaokao2023en | 57.92 | 65.20 | 64.16 | 62.08 |
| | Minerva Math | 34.93 | 38.24 | 36.77 | 33.82 |
| | Olympiad Bench | 42.58 | 52.34 | 50.00 | 42.77 |
| | Average | 51.97 | 58.43 | 56.30 | 52.77 |

### H.3 MORE ABLATION STUDY

In this section, we provide an extended ablation study to evaluate the contribution of each key component in Dr. MAMR. Table 6 summarizes the results for the 7B model across several math-reasoning benchmarks. These ablations are designed specifically to address the reviewer's questions regarding (1) the effectiveness of the proposed Shapley-inspired grouping mechanism, (2) the choice of the cosine-similarity threshold for grouping, and (3) the robustness of the causal-influence (CI) computation to different embedding models.

(1) Effectiveness of Shapley-inspired grouping:

A core part of Dr. MAMR is the Shapley-inspired semantic grouping, which clusters intermediate reasoning segments into conceptually meaningful groups before computing causal influence. To assess its importance, we report a w/o Shapley in CI variant, where grouping is removed and CI is computed directly on every segment independently. As shown in Table 6, removing Shapley grouping worsens performance across all benchmarks (e.g., AIME24: 20.00 → 16.67, AMC23: 62.50 → 60.00). This confirms that grouping reduces noise in the CI estimates by aggregating semanti-

cally redundant reasoning segments, and that the Shapley mechanism meaningfully contributes to the stability and effectiveness of the causal-influence reward.

(2) Varying the semantic-similarity threshold:

Dr. MAMR uses a cosine-similarity threshold of 0.9 to decide whether two reasoning segments belong to the same group. To evaluate sensitivity to this design choice, we vary the threshold to 0.95 and 0.8. Results in Table 6 show: (1) 0.95 (more restrictive grouping): performance remains similar or slightly improved on some datasets, indicating that Dr. MAMR is stable when grouping is more conservative. (2) 0.8 (more aggressive grouping): performance drops more noticeably, suggesting that overly broad grouping merges semantically distinct segments, which weakens CI's ability to identify the truly influential steps.

Overall, Dr. MAMR maintains strong performance for a wide range of thresholds, demonstrating robustness, but extremely permissive grouping can blur important distinctions in the reasoning trajectory.

(3) Effect of embedding model used for grouping:

By default, semantic grouping uses Qwen2.5-0.5B embeddings. To test robustness to this choice, we replace the embeddings with: (1) Qwen2-0.5B, a smaller instruction-tuned model (2) Qwen3-Embedding-0.6B, a specialized embedding model

As shown in Table 6, the method remains stable across embedding variants. Notably, Qwen3-Embedding-0.6B improves grouping quality (e.g., Gaokao2023en: $65.20 \rightarrow 66.49$, Olympiad: $52.34 \rightarrow 54.52$), likely because its representations better capture semantic similarity. These results indicate that CI is robust to embedding model choice, and improvements can be obtained with stronger embedding encoders.

Table 6: Ablation study on the 7B model.

| Variant | AIME24 | AMC23 | Gaokao2023en | Olympiad Bench |
|---|---|---|---|---|
| Dr.MAMR | 20.00 | 62.50 | 65.20 | 52.34 |
| w/o ND | 13.33 | 55.00 | 63.64 | 47.85 |
| w/o CI | 13.33 | 52.50 | 63.38 | 45.31 |
| w/o RB | 16.67 | 57.50 | 63.90 | 50.58 |
| w/o Shapley in CI | 16.67 | 60.00 | 64.42 | 49.22 |
| w/ Qwen2-0.5B for CI | 16.67 | 60.00 | 63.12 | 51.56 |
| w/ Qwen3-Embedding-0.6B for CI | 20.00 | 62.50 | 66.49 | 54.52 |
| w/ threshold 0.95 for CI | 20.00 | 60.00 | 65.71 | 52.73 |
| w/ threshold 0.8 for CI | 16.67 | 57.50 | 62.86 | 50.20 |

Table 7: Comparison to stronger single-agent RL baseline.

| Model | Benchmark | GRPO | DAPO | Dr. GRPO | GSPO | Dr. MAMR |
|---|---|---|---|---|---|---|
| Qwen2.5 -7B -Instruct | MATH500 | 75.50 | 75.80 | 75.60 | 76.20 | 78.60 |
| | GSM8K | 90.50 | 93.26 | 91.13 | 91.05 | 92.12 |
| | AIME24 | 16.67 | 13.33 | 16.67 | 16.67 | 20.00 |
| | AMC23 | 55.00 | 60.00 | 52.50 | 57.50 | 62.50 |
| | Gaokao2023en | 64.60 | 63.64 | 64.42 | 65.71 | 65.20 |
| | Minerva Math | 34.70 | 35.60 | 35.30 | 37.50 | 38.24 |
| | Olympiad Bench | 48.60 | 49.61 | 47.07 | 50.78 | 52.34 |
| | Average | 55.08 | 55.89 | 54.67 | 56.49 | 58.43 |

## H.4 RESULTS ON CODE BENCHMARK

To evaluate the generalization capability of Dr. MAMR beyond math reasoning, we conduct experiments on code generation benchmarks. We use the DeepCoder dataset[7] for training and adopt DeepSeek-R1-Distill-Qwen-1.5B as the base model. We compare Dr. MAMR against two baselines: the single-agent GRPO and the multi-agent ReMA, evaluating performance on three widely used code reasoning benchmarks: LiveCodeBench, Codeforces, and HumanEval+. For each benchmark, we report pass@k metrics at $k \in 1, 2, 4, 8$.

As shown in Tables 10, 8, and 9, Dr. MAMR consistently outperforms both GRPO and ReMA across all benchmarks and almost all pass@$k$ settings. On LiveCodeBench, Dr. MAMR achieves a substantial improvement in pass@1 (19.35%) compared to GRPO (15.77%) and ReMA (15.05%). Similar gains are observed on HumanEval+ (61.35% vs. 58.28%/57.67%). These results demonstrate that Dr. MAMR not only excels in math reasoning but also generalizes effectively to complex code-based reasoning tasks, validating its robustness across diverse domains.

We further visualize the training dynamics of Dr. MAMR compared to ReMA in Figure 8. The plotted curve reports the mean reward throughout the RL training process. We observe that while ReMA's performance quickly plateaus and deteriorates—ultimately collapsing to near-zero values—Dr. MAMR demonstrates sustained and progressively improving performance. This indicates that Dr. MAMR enables stable optimization in long-horizon multi-agent reasoning tasks.

Table 8: Comparison on the Codeforces benchmark (pass@k).

| Metric | GRPO | ReMA | Dr. MAMR |
|--------|------|------|----------|
| pass@1 | 5.88 | 4.66 | 6.22 |
| pass@2 | 8.33 | 7.35 | 8.33 |
| pass@4 | 11.03 | 11.27 | 12.25 |
| pass@8 | 15.44 | 14.95 | 15.93 |

Table 9: Comparison on the HumanEval+ benchmark (pass@k).

| Metric | GRPO | ReMA | Dr. MAMR |
|--------|------|------|----------|
| pass@1 | 58.28 | 57.67 | 61.35 |
| pass@2 | 69.33 | 70.55 | 71.78 |
| pass@4 | 77.30 | 76.69 | 79.14 |
| pass@8 | 83.44 | 82.82 | 84.66 |

Table 10: Comparison on the LiveCodeBench benchmark (pass@k)

| Metric | GRPO | ReMA | Dr. MAMR |
|--------|------|------|----------|
| pass@1 | 15.77 | 15.05 | 19.35 |
| pass@2 | 19.35 | 17.56 | 22.58 |
| pass@4 | 25.45 | 22.22 | 26.88 |
| pass@8 | 29.39 | 24.37 | 32.26 |

---

[7]agentica-org/DeepCoder-Preview-Dataset

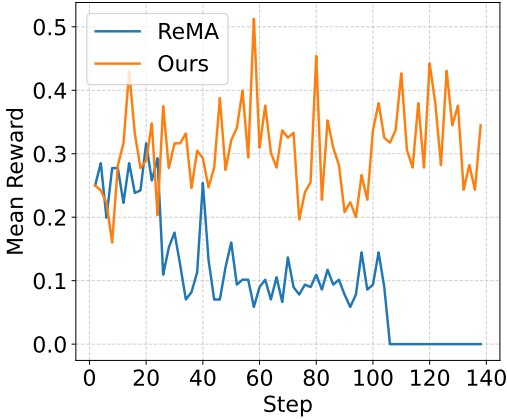

Figure 8: Training curves comparison on code dataset.

---

**System Prompts for Meta-Think and Reasoning Agents (Coding)**

**META-THINK AGENT SYSTEM PROMPT**
You are a meta-think agent that represents human high-level thought process when solving coding problems. You will have a discussion with the reasoning agent. Each time you think about what to do next, consider:

- Analyzing the problem requirements and constraints carefully
- Breaking down the solution into clear algorithmic steps
- Identifying key data structures and algorithms needed
- Considering edge cases and potential bugs
- Requesting implementation of specific functions or code blocks
- Reviewing intermediate code and suggesting improvements
- Backtracking when the approach doesn't work
- Finally confirm when the solution is complete with the tag `[FINISH]`

**REASONING AGENT SYSTEM PROMPT**
You are a reasoning agent that implements code solutions step by step following the meta-think agent's instructions. When writing code:

- Implement the solution clearly and correctly
- Include proper error handling for edge cases
- Use appropriate data structures and algorithms
- Write clean, readable code with proper formatting
- When asked to finalize your answer, put your complete code solution within a markdown code block using triple backticks (` ```python ... ``` `)
- Make sure your final code is complete and can be executed directly

---

## H.5 TRAINING TIME COMPARISON

We report the per-step training time of Dr. MAMR compared to GRPO, with results shown in Figure 9. As illustrated in the figure, the training time of Dr. MAMR is generally comparable to that of the single-agent GRPO baseline.

## H.6 RESULTS ACROSS BENCHMARKS ON 3B MODEL

In this section, we present additional pass@1 performance results on the 3B model, as shown in Table 11. From the table, we observe that our Dr. MAMR consistently outperforms both single-agent

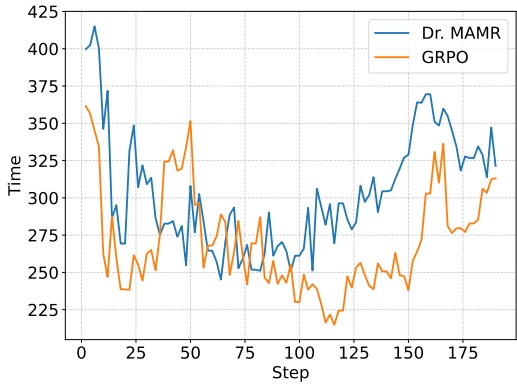

Figure 9: Per-step training time comparison.

GRPO and ReMA. However, the performance gains are less pronounced on the 7B and 14B models. We attribute this to their weaker instruction-following capability, which limits the performance upper bound of the multi-agent system.

Table 11: Performance on math benchmarks with 3B base model.

| Model | Benchmark | GRPO | VRP (CoT) | ReMA | Dr. MAMR (Ours) |
|---|---|---|---|---|---|
| | MATH500 | 65.60 | 65.20 | 62.60 | 66.20 |
| | GSM8K | 85.30 | 72.02 | 83.17 | 85.37 |
| | AIME24 | 13.33 | 3.33 | 3.33 | 16.67 |
| Qwen2.5 -3B -Instruct | AMC23 | 40.00 | 20.00 | 42.50 | 50.00 |
| | Gaokao2023en | 54.30 | 30.91 | 52.73 | 55.33 |
| | Minerva Math | 31.20 | 16.91 | 26.47 | 32.35 |
| | Olympiad Bench | 30.20 | 6.07 | 27.56 | 30.57 |
| | **Average** | 45.70 | 30.63 | 42.62 | 48.07 |

### H.7 TRAINING CURVE COMPARISON ON THE 3B MODEL

To further examine the training stability of smaller base models, we provide additional experiments on the 3B model. In this setting, we use MATH as the training dataset to rule out the possibility that collapse is solely caused by overly difficult data. The training curves reporting mean reward are shown in Fig. 10.

From the figure, we observe that under the ReMA framework, the 3B model collapses rapidly after only 20 training steps. In contrast, our Dr. MAMR framework maintains stable progress throughout training. This highlights how critical the lazy-agent issue becomes when the base model is relatively weak. Importantly, even with a less capable model, Dr. MAMR is still able to ensure stable training, underscoring its robustness.

### H.8 CASE STUDY ON RESTART BEHAVIOR

In this section, we present a case study on restart behavior, illustrating how restarts help the reasoning agent recover from its earlier mistakes. The full reasoning process is shown below. This case study demonstrates how restart behavior enables the agent to abandon an incomplete reasoning trajectory and reinitiate a more systematic approach. By explicitly reconsidering the intersection points, the agent successfully recovered the correct quadrilateral and computed the ground-truth area of $4\sqrt{3}$. Without such a restart, the agent would likely have produced an incorrect or incomplete solution.

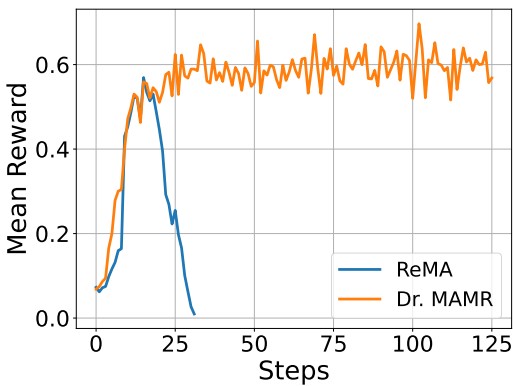

Figure 10: Training curves of 3B model trained with ReMA and Dr. MAMR.

---

**Case study on restart behavior**

[Question] Find the area of the quadrilateral formed by the intersection points of the parabolas $y^2 = 3x$, $y^2 = -3x$, and the circle $x^2 + y^2 = 4$. **[Groundtruth: $4\sqrt{3}$]**

**Meta-thinking:**

To find the area of the quadrilateral formed by the points of intersection between the parabolas and the circle, begin by determining the intersection points. Substitute $y^2 = 3x$ and $y^2 = -3x$ into $x^2 + y^2 = 4$ and work through the feasible solutions.

**Reasoning:**

For $y^2 = 3x$: $x^2 + 3x = 4 \implies (x+4)(x-1) = 0$, so $x = 1$ or $-4$. But $x = -4$ gives $y^2 = -12$. Thus $(1, \pm\sqrt{3})$.

For $y^2 = -3x$: $x^2 - 3x = 4 \implies (x-4)(x+1) = 0$, so $x = 4$ or $-1$. But $x = 4$ gives $y^2 = -12$. Thus $(-1, \pm\sqrt{3})$.

**Meta-thinking:**

The feasible intersections suggest four symmetric points on the circle: $(1, \pm\sqrt{3})$ and $(-1, \pm\sqrt{3})$. It's natural to check the configuration these points form and then evaluate the area accordingly.

**Reasoning:**

<restart> Decision: Rethink from scratch. The feasible intersections are: - From $y^2 = 3x$: $(1, \pm\sqrt{3})$ - From $y^2 = -3x$: $(-1, \pm\sqrt{3})$

Thus, four points of intersection: $A = (1, \sqrt{3}), B = (1, -\sqrt{3}), C = (-1, \sqrt{3}), D = (-1, -\sqrt{3})$.

**Meta-thinking:**

These points form a rectangle. - Width: $|1 - (-1)| = 2$ - Height: $|\sqrt{3} - (-\sqrt{3})| = 2\sqrt{3}$

Area:

$$\text{Area} = 2 \times 2\sqrt{3} = 4\sqrt{3}$$

[FINISH]
**Reasoning:**

$$\boxed{4\sqrt{3}}$$

---

## H.9 SCALING

In this section, we present the full results of test-time scaling on pass@K performance when comparing Dr. MAMR with ReMA, as shown in Fig. H.9. We observe that on harder benchmarks such as AIME24 and OlympiadBench, the performance gap between Dr. MAMR and ReMA widens as $K$ increases, while on relatively easier benchmarks, Dr. MAMR consistently achieves better performance. These results highlight the strong capability of Dr. MAMR in handling a wide range of reasoning tasks.

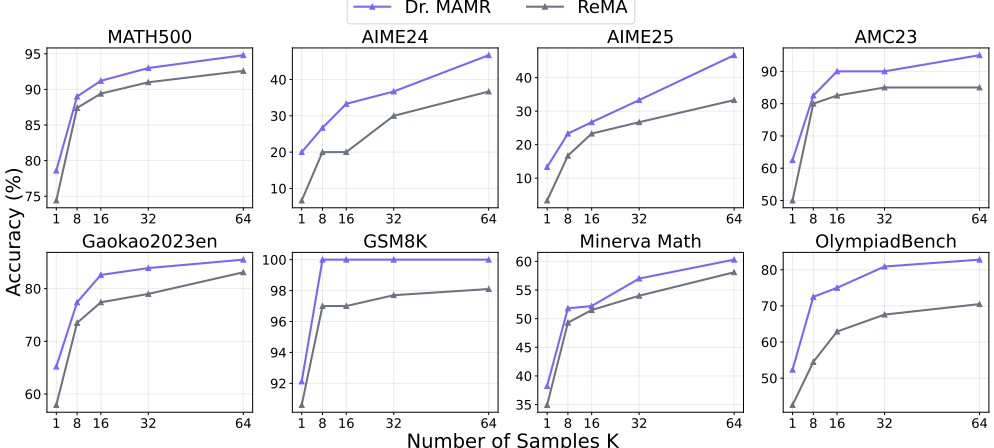

Figure 11: Pass@K performance.

# I  THE USE OF LARGE LANGUAGE MODELS

Large Language Models (LLMs) were used to assist in writing and polishing this manuscript. Specifically, an LLM was employed to improve clarity, refine language, check grammar, and enhance overall readability.

The LLM was not involved in ideation, research design, data analysis, or the development of scientific content. All research concepts, methods, and analyses were independently conceived and conducted by the authors.

The authors take full responsibility for the manuscript's content, including any text refined with LLM assistance. All LLM-generated content adheres to ethical standards and does not constitute plagiarism or scientific misconduct.

