# OpenReview forum: "Unlocking the Power of Multi-Agent LLM for Reasoning: From Lazy Agents to Deliberation"
_ICLR.cc/2026/Conference — ICLR 2026 Poster_

### Official Review · Reviewer_eVXe · 2025-10-28

**Soundness:** 3
**Presentation:** 3
**Contribution:** 3
**Rating:** 6
**Confidence:** 3

**Summary:**

This paper proposes DR MAMR, a series of enhancements for reinforcement learning algorithms focused on multi-turn multi-agent systems like those introduced in ReMA (Wan 2025). Here, the focus is on multi-agent systems that alternate between a meta-agent, whose purpose is to provide higher-level planning and form intermediate tasks, and a reasoner agent, whose purpose is to reason about and solve tasks proposed by the meta-agent. DR MAMR identifies a “lazy agent” shortcoming in ReMA, where a single agent does all of the planning and reasoning, while the other agent produces short (or no) outputs. The authors then propose fixes to this involving modifications to the loss function and methods for provider denser credit attribution via Shapely values. The authors train models at 3B, 7B, and 14B scale and evaluate on math tasks.

**Strengths:**

- Shapely values are an interesting, and to my knowledge, novel way to provide dense credit attribution in multi-turn settings with relatively small computational overhead (a 0.5B embedding model is required to compute semantic similarity). The ablation result shows that such credit attribution has a large impact on performance.
- The proposed changes and additions to the loss and reward formulation are principle and experimentally backed with controlled ablations.
- The paper is generally well-written and the problem of the lazy agent is well-motivated

**Weaknesses:**

- Relatively weak baseline choices: The second performing baseline across model scales is GRPO training. There exist many algorithmic improvements on top of GRPO that enable better performance, such as DAPO, DR GRPO. The advantage of these methods is that they do not require such complicated rollout and reward approaches needed by ReMA and DR MAMR. Even a comparison across one model size, like 7B, would be helpful.
- The authors focus only on math domains, eschewing other reasoning domains/benchmarks. As a starting point, I would love to see if DR MAMR models generalize to other domains, e.g., evaluating on GPQA-Diamond, MMLU-Pro, etc.
- Theoretical presentation: The proposed theorem is a trivial re-arrangement of the defined $g, Z$ terms. The full proof of this result reveals the simplicity, as it spends the majority of the space deriving the expression for $Z$ (by simply performing some algebraic manipulations involving gradient computations), then concluding with some additional algebraic manipulations. Overall, I find this result overstated: It does not introduce any novel result, nor does it use novel techniques in the proof. Presenting such a result as a standalone theorem needlessly complicates presentation.

**Questions:**

- Do authors compare Shapely-valued dense rewards with alternative methods, like using a process reward model?
- Nit: GRPO stands for Group Relative Policy Optimization, not Preference Optimization (L46)

---

> ### Author Response · Authors · 2025-11-23
>
> We sincerely thank the reviewer for acknowledging that our idea is interesting and that our method is well-motivated. Below, we provide a detailed response to each concern.
>
> ------
>
> >###  W1: Relatively weak baseline choices
>
> We thank the reviewer for raising this important point. We agree that it is valuable to include GRPO-based improvements as baselines. Following your suggestion, we conducted additional experiments under the same setup as Sec. 6.1 and compared **Dr. MAMR** against **DAPO**, **Dr. GRPO**, and **GSPO**.
>
> The results are presented in the table below. While DAPO and GSPO do improve the performance of single-agent GRPO, Dr. MAMR continues to outperform all GRPO-based variants across benchmarks, demonstrating the strong potential of multi-agent RL—even when compared against stronger single-agent baselines.
>
> | Model                  | Benchmark       | GRPO  | DAPO  | Dr. GRPO | GSPO  | Dr. MAMR |
> |------------------------|-----------------|--------|--------|----------|--------|----------|
> | Qwen2.5-7B-Instruct    | MATH500         | 75.50  | 75.80  | 75.60    | 76.20  | 78.60    |
> |                  | GSM8K           | 90.50  | 93.26  | 91.13    | 91.05  | 92.12    |
> |              | AIME24          | 16.67  | 13.33  | 16.67    | 16.67  | 20.00    |
> |               | AMC23           | 55.00  | 60.00  | 52.50    | 57.50  | 62.50    |
> |                | Gaokao2023en    | 64.60  | 63.64  | 64.42    | 65.71  | 65.20    |
> |                | Minerva Math    | 34.70  | 35.60  | 35.30    | 37.50  | 38.24    |
> |               | Olympiad Bench  | 48.60  | 49.61  | 47.07    | 50.78  | 52.34    |
> |                | **Average**     | 55.08  | 55.89  | 54.67    | 56.49  | 58.43    |
>
> ---------
>
> >###   W2: focus only on math domains &  generalize to other domains, e.g., evaluating on GPQA-Diamond, MMLU-Pro, etc.
>
> >### **(1) experiments on code benchmark**
>
> We thank the reviewer for raising this important point. To evaluate the generalization capability of Dr. MAMR beyond math reasoning, we conduct experiments on code generation benchmarks. We use the DeepCoder dataset for training and adopt DeepSeek-R1-Distill-Qwen-1.5B as the base model. We compare Dr. MAMR against two baselines: the single-agent GRPO and the multi-agent ReMA, evaluating performance on three widely used code reasoning benchmarks: LiveCodeBench, Codeforces, and HumanEval+. For each benchmark, we report pass@k metrics at $k \in {1, 2, 4, 8}$. More details and prompt can be found in H.5 in Appendix. We report results in table below:
>
>
> |Benchmark| Metric  | GRPO | ReMA | Dr. MAMR |
> |----|---------|------|------|-----------|
> |Codeforces| pass@1  | 5.88 | 4.66 | 6.22  |
> || pass@2  | 8.33 | 7.35 | 8.33  |
> || pass@4  | 11.03| 11.27| 12.25 |
> || pass@8  | 15.44| 14.95| 15.93 |
>
> |Benchmark| Metric  | GRPO  | ReMA  | Dr. MAMR |
> |----|---------|-------|-------|-----------|
> |HumanEval+| pass@1  | 58.28 | 57.67 | 61.35 |
> || pass@2  | 69.33 | 70.55 | 71.78 |
> || pass@4  | 77.30 | 76.69 | 79.14 |
> || pass@8  | 83.44 | 82.82 | 84.66 |
>
>
> |Benchmark| Metric  | GRPO  | ReMA  | Dr. MAMR |
> |-----|---------|-------|-------|-----------|
> |LiveCodeBench| pass@1  | 15.77 | 15.05 | 19.35 |
> || pass@2  | 19.35 | 17.56 | 22.58 |
> || pass@4  | 25.45 | 22.22 | 26.88 |
> || pass@8  | 29.39 | 24.37 | 32.26 |
>
>
>
> As shown in above tables, Dr. MAMR consistently outperforms both GRPO and ReMA across all benchmarks and almost all pass@$k$ settings. On LiveCodeBench, Dr. MAMR achieves a substantial improvement in pass@1 (19.35\%) compared to GRPO (15.77\%) and ReMA (15.05\%). Similar gains are observed on HumanEval+ (61.35\% vs. 58.28\%/57.67\%). These results demonstrate that Dr. MAMR not only excels in math reasoning but also generalizes effectively to complex code-based reasoning tasks, validating its robustness across diverse domains.
>
>
>
> **We further visualize the training dynamics of Dr. MAMR compared to ReMA in Figure 8 in the appendix**. The plotted curve reports the mean reward throughout the RL training process. We observe that while ReMA’s performance quickly plateaus and deteriorates—ultimately collapsing to near-zero values—Dr. MAMR demonstrates sustained and progressively improving performance. This indicates that Dr. MAMR enables stable optimization in long-horizon multi-agent reasoning tasks.
>
> >### **(2) evaluation on GPQA-Diamond, MMLU-Pro**
>
> We appreciate the reviewer’s suggestion. To assess whether Dr. MAMR’s reasoning ability generalizes beyond math, we further evaluate the model (trained on math datasets) on **GPQA-Diamond** and **MMLU-Pro**. Using Qwen2.5-7B-Instruct as the base model, the results are shown below.
>
> |Model| MMLU_pro |GPQA_Diamond  |
> | ---|--|--|
> |Base | 56.3 | 33.8 |
> |Dr. MAMR |58.3 |40.4 |
>
> These results show that **Dr. MAMR** improves general reasoning performance across domains, suggesting that the cooperative multi-agent training enhances reasoning transferability beyond the math domain.

---

> ### Author Response · Authors · 2025-11-23
>
> >###   W3: Theoretical presentation
>
>
> We sincerely appreciate the reviewer for carefully examining our theoretical section and for the constructive feedback.
>
> Our idea is that when realistic reward noise is taken into consideration, it becomes difficult to quantitatively compare the exact gradient magnitudes of two trajectories, and thus we intentionally abstract away this confounder. The goal is to demonstrate that although noise makes the comparison less precisely measurable, it does not change the underlying directional preference: shorter trajectories systematically receive up-weighted gradient updates.
>
> We appreciate the reviewer highlighting this, and in future revisions we will present the intuition more directly and reduce the technical overhead so the main idea is clearer.
>
> ----------
>
> >###   Q1: Do authors compare Shapely-valued dense rewards with alternative methods, like using a process reward model?
>
> We sincerely thank the reviewer for this insightful suggestion. We did conduct an experiment using a process reward model, as reported in **Appendix G**, and apologize for not referencing it clearly in the main text. In this experiment, we followed the setup in Section 6.1 and adopted **Qwen/Qwen2.5-Math-PRM-7B** as the process reward model (PRM), adding its process reward to each turn’s reward.
>
> As shown in **Fig. 6 (Appendix)**, this approach collapses rapidly within 30 training steps, which we attribute to **reward hacking**—a common issue when dense process rewards directly evaluate turn-level quality. In contrast, our Shapley-style causal-influence estimator focuses only on each turn’s contribution to the next reasoning step and leaves quality evaluation to the final verifiable reward, thereby avoiding reward hacking and providing more stable credit signals.
>
> ---------
>
> >###   Q2: GRPO stands for Group Relative Policy Optimization, not Preference Optimization (L46)
>
> We sincerely thank the reviewer for carefully reading our paper and for catching this mistake. We have fixed it.

---

> > ### Comment · Reviewer_eVXe · 2025-11-27
> >
> > Thanks to the authors for their hard work in addressing comments. My concerns have been largely addressed. It's great to see Dr MAMR brings gains over other algorithmic changes, and that gains generalize to other domains and models. I'll update my score.

---

> > > ### Author Response · Authors · 2025-11-27
> > >
> > > Dear Reviewer eVXe:
> > >
> > > We sincerely thank the reviewer for the thoughtful follow-up, the constructive feedback, and for increasing the score after reviewing our additional experiments and clarifications. Your engagement has meaningfully strengthened the paper, and we truly appreciate your support. Thank you very much!
> > >
> > > Best,
> > >
> > > Submission #14685 Authors

---

### Official Review · Reviewer_7UcQ · 2025-10-31

**Soundness:** 3
**Presentation:** 3
**Contribution:** 3
**Rating:** 6
**Confidence:** 4

**Summary:**

The paper studies why lazy-agent behavior emerges in multi‑agent LLM reasoning (one agent effectively does all the work while the other contributes little) and proposes a theoretical framework to address this issue by tracing the problem to a bias in multi‑turn GRPO: the turn‑length normalization favors shorter trajectories, implicitly discouraging collaborative back‑and‑forth. Then introduce Dr. MAMR with three components: (i) remove the turn‑length normalization identified by the theory; (ii) add a Shapley‑inspired step‑level causal‑influence signal computed by grouping semantically similar steps across rollouts(which is common is MARL setting); and (iii) add a verifiable reward for restart via a <restart> control that lets the reasoning agent discard noisy history and begin afresh. On seven math benchmarks with Qwen2.5 7B and 14B‑Instruct, Dr. MAMR outperforms single‑agent GRPO, VRP(CoT), and ReMA. And improves stability (ReMA collapses; Dr. MAMR does not).

**Strengths:**

-- Motivation: The paper motivates that lazy‑agent behavior collapses multi‑agent systems into single‑agent reasoning, squandering collaboration benefits, which is an issue observed in MARL and newly shown here in LLM multi‑agent reasoning. The motivation is explicit in the introduction and contribution summary.

Well structured:

-- Crisp theoretical diagnosis of the cause. Theorem 1 in (Sec. 5.1, p. 5) formalizes how the 1/T normalization in multi‑turn GRPO biases the gradient toward shorter (fewer‑turn) continuations with equal final reward, explaining the emergence of lazy agents; the authors also argue the bias is pronounced because turns are far fewer than tokens. (Sec. 5.1, p. 5)

-- Methodologically well‑scoped fix. The Shapley‑inspired causal‑influence estimator averages masked‑history log‑likelihood deltas over groups of semantically similar steps, reducing phrasing bias and variance. A “who helped whom” question. (Sec. 5.2, p. 6; Appx. B.3, p. 18)

**Weaknesses:**

-- Domain generality is narrow. All experiments are on math; claims about “complex reasoning tasks” would be stronger with code, planning, or QA tasks where verifiability is harder. Which can lead to another question:
-- Restart relies on verifiable end‑state. The restart reward needs a checkable final answer; outside math (or without ground‑truth/validators), the mechanism may not be directly applicable. Moreover, ablations show restart helps but less than CI/normalization fixes, suggesting narrower impact. (Sec. 5.3 p. 7; Table 2 p. 9)

-- Theory relies on an equal final reward assumption. Theorem 1 contrasts continuations with equal final reward but different lengths; this is a clean but restrictive setting. It is less clear how strongly the conclusion holds under more realistic reward noise, token‑level normalizations/clipping, or different credit schemes. (Sec. 5.1, p. 5)

-- Sensitivity of the CI estimator is underexplored. Grouping uses Qwen2.5‑0.5B embeddings and a cosine‑similarity threshold of 0.9; \alpha = \beta = 0.1 are fixed across runs for stability. The paper lacks ablations on embedding choice/threshold and \alpha /\beta, so robustness of CI to these design choices is uncertain. (Appx. B.2–B.3, p. 18)

-- Connection to MARL: Comparative context to prior “lazy‑agent” literature could be tighter. The paper cites MARL work on lazy agents but does not empirically compare to mechanisms from that line (e.g., influence‑based credit or external‑state influence). This limits continuity with prior method. (Sec. 2, p. 3; cf. ICML’23 “Lazy Agents” & Lazy‑MDPs)

**Questions:**

Besides the weakness, I have something to ask:

-- Theory stress tests: Beyond removing 1/T, did you try alternatives (e.g., adaptive normalization by effective turns that exclude no‑ops, or per‑role normalization)? Do the theoretical predictions hold when rewards differ slightly (near‑equal) or when token‑level normalizations are active? (Sec. 5.1, p. 5)

-- The phenomenon of “lazy agents” has been studied in MARL (e.g., LAIES at ICML’23; Lazy‑MDPs). This paper’s contribution is to diagnose and mitigate it in multi‑agent LLM reasoning with a targeted loss fix, a tractable step‑level influence signal, and a restart‑credit mechanism. What's your next plan for further work to combine this work?

---

> ### Author Response · Authors · 2025-11-23
> **Response to W1**
>
> We thank the reviewer for acknowledging the clarity of our motivation and the crispness of our theoretical diagnosis. Below, we provide a point-by-point response to the reviewer’s comments:
>
> >### W1: Domain generality is narrow. All experiments are on math; claims about “complex reasoning tasks” would be stronger with code, planning, or QA tasks where verifiability is harder. Which can lead to another question: -- Restart relies on verifiable end‑state. The restart reward needs a checkable final answer; outside math (or without ground‑truth/validators), the mechanism may not be directly applicable.
>
> >### **(1) Domain generality is narrow**
>
> To evaluate the generalization capability of Dr. MAMR beyond math reasoning, we conduct experiments on code generation benchmarks. We use the DeepCoder dataset for training and adopt DeepSeek-R1-Distill-Qwen-1.5B as the base model. We compare Dr. MAMR against two baselines: the single-agent GRPO and the multi-agent ReMA, evaluating performance on three widely used code reasoning benchmarks: LiveCodeBench, Codeforces, and HumanEval+. For each benchmark, we report pass@k metrics at $k \in {1, 2, 4, 8}$. More details and prompt can be found in H.5 in Appendix. We report results in table below:
>
>
> |Benchmark| Metric  | GRPO | ReMA | Dr. MAMR |
> |----|---------|------|------|-----------|
> |Codeforces| pass@1  | 5.88 | 4.66 | 6.22  |
> || pass@2  | 8.33 | 7.35 | 8.33  |
> || pass@4  | 11.03| 11.27| 12.25 |
> || pass@8  | 15.44| 14.95| 15.93 |
>
>
>
> |Benchmark| Metric  | GRPO  | ReMA  | Dr. MAMR |
> |----|---------|-------|-------|-----------|
> |HumanEval+| pass@1  | 58.28 | 57.67 | 61.35 |
> || pass@2  | 69.33 | 70.55 | 71.78 |
> || pass@4  | 77.30 | 76.69 | 79.14 |
> || pass@8  | 83.44 | 82.82 | 84.66 |
>
>
>
> |Benchmark| Metric  | GRPO  | ReMA  | Dr. MAMR |
> |-----|---------|-------|-------|-----------|
> |LiveCodeBench| pass@1  | 15.77 | 15.05 | 19.35 |
> || pass@2  | 19.35 | 17.56 | 22.58 |
> || pass@4  | 25.45 | 22.22 | 26.88 |
> || pass@8  | 29.39 | 24.37 | 32.26 |
>
>
>
> As shown in above tables, Dr. MAMR consistently outperforms both GRPO and ReMA across all benchmarks and almost all pass@$k$ settings. On LiveCodeBench, Dr. MAMR achieves a substantial improvement in pass@1 (19.35\%) compared to GRPO (15.77\%) and ReMA (15.05\%). Similar gains are observed on HumanEval+ (61.35\% vs. 58.28\%/57.67\%). These results demonstrate that Dr. MAMR not only excels in math reasoning but also generalizes effectively to complex code-based reasoning tasks, validating its robustness across diverse domains.
>
>
>
> **We further visualize the training dynamics of Dr. MAMR compared to ReMA in Figure 8 in the appendix**. The plotted curve reports the mean reward throughout the RL training process. We observe that while ReMA’s performance quickly plateaus and deteriorates—ultimately collapsing to near-zero values—Dr. MAMR demonstrates sustained and progressively improving performance. This indicates that Dr. MAMR enables stable optimization in long-horizon multi-agent reasoning tasks.
>
>
>
>
>
> >### **(2) Restart relies on verifiable end‑state. The restart reward needs a checkable final answer; outside math (or without ground‑truth/validators), the mechanism may not be directly applicable.**
>
> We thank the reviewer for raising this important point. Our work is situated within the rapidly growing line of reinforcement learning with verifiable rewards, where the final output can be checked deterministically. However, when a deterministic validator is not available, a natural extension of our restart mechanism still applies.
>
> If the reasoning agent outputs restart at turn _t_, we can generate a **counterfactual non-restart continuation** for the same prompt (i.e., force the agent to continue instead of restarting). Both final answers can then be scored by the same reward function—which is already required in all RL settings, including non-verifiable tasks. The restart reward is defined as the improvement in reward-model score obtained by restarting relative to this counterfactual continuation.
>
> This preserves the core principle of our design: **restart is rewarded only when it truly improves the final answer**, regardless of whether the evaluator is a deterministic math verifier or a learned reward model. Therefore, the mechanism naturally generalizes beyond math and remains applicable to any RL setting where a reward model or evaluator is available.

---

> ### Author Response · Authors · 2025-11-23
> **Response to W1(2)**
>
> >### **(3) ablations show restart helps but less than CI/normalization fixes, suggesting narrower impact.**
>
> We thank the reviewer for the insightful observation. It is true that the restart mechanism contributes a smaller improvement than CI and the multi-turn normalization fix. This is expected because CI and normalization address fundamental structural issues—they directly correct how each agent contributes to the final result and therefore have a larger impact on overall performance. In contrast, the restart mechanism targets a specific failure mode: when the reasoning agent becomes lost in long multi-turn trajectories.
>
>
> Despite this, restart still provides meaningful gains—for example, a 5% improvement on AMC23 in Table 2. To fully address your concern and illustrate the role of restart, we conducted an additional ablation under Pass@K evaluation for AIME24 and AIME25.
>
>
>
> || Model | pass@1 | pass@8 | pass@16 | pass@32 | pass@64 |
> |----| ---------- | ------ | ------ | ------- | ------- | ------- |
> |AIME24|  **Dr. MAMR**  | 20.00 | 26.67 | 33.33 | 36.67 | 46.67 |
> ||  **Dr. MAMR/restart**  | 16.67 | 23.33 | 30.00 | 33.33 | 40.00 |
>
>
>
> || Model | pass@1 | pass@8 | pass@16 | pass@32 | pass@64 |
> |----| ---------- | ------ | ------ | ------- | ------- | ------- |
> |AIME25|  **Dr. MAMR**  | 13.33 | 23.33 | 26.67 | 33.33 | 46.67 |
> ||  **Dr. MAMR/restart**  | 10.00 | 20.00 | 23.33 | 30.00 | 40.00 |
>
>
>
> Across both benchmarks, the gap shows a upward trend as more attempts are allowed. This shows that while restart is not the primary driver of overall gains, it provides important complementary benefit by improving stability in long-horizon reasoning, especially when multiple solution attempts are permitted.

---

> ### Author Response · Authors · 2025-11-23
> **Response to W2 & W3 (1)**
>
> >### W2: Theory relies on an equal final reward assumption. Theorem 1 contrasts continuations with equal final reward but different lengths; this is a clean but restrictive setting. It is less clear how strongly the conclusion holds under more realistic reward noise, token‑level normalizations/clipping, or different credit schemes. (Sec. 5.1, p. 5)
>
>
>
>
> We appreciate the reviewer’s concern regarding the equal-final-reward assumption in Theorem 1. Indeed, when realistic reward noise is taken into consideration, it becomes difficult to quantitatively compare the exact gradient magnitudes of two trajectories, and our theorem intentionally abstracts away this confounder. However, the presence of noise does not alter the structural bias induced by the 1/T normalization in multi-turn GRPO. Regardless of whether rewards are noisy, clipped, or estimated with variance, the update for a long trajectory is still scaled by $1/T_L$, while the update for a shorter trajectory is scaled by $1/T_S$. Thus, **although noise makes the comparison less precisely measurable, it does not change the underlying directional preference: shorter trajectories systematically receive upweighted gradient updates.**
>
>
>
> This is exactly what we observe in practice. In the full, noisy training pipeline—where final rewards differ, advantages fluctuate, and token-level clipping is applied—the reasoning agent collapses (Fig. 4), showing that the length-induced bias persists under realistic conditions. Noise perturbs the gradients but does not remove the normalization effect;
>
>
>
>
> >### W3: Sensitivity of the CI estimator is underexplored. Grouping uses Qwen2.5‑0.5B embeddings and a cosine‑similarity threshold of 0.9; \alpha = \beta = 0.1 are fixed across runs for stability. The paper lacks ablations on embedding choice/threshold and \alpha /\beta, so robustness of CI to these design choices is uncertain. (Appx. B.2–B.3, p. 18)
>
> >### **(1) ablation study on threshold and embedding model in CI estimator**
>
> We thank the reviewer for pointing this out. To address your concern, we provide an extended ablation study.
> - (i) **Varying the semantic-similarity threshold**: Dr. MAMR uses a cosine-similarity threshold of 0.9 to decide whether two reasoning segments belong to the same group. To evaluate sensitivity to this design choice, we vary the threshold to 0.95 and 0.8. Results in Table below show: (1) 0.95 (more restrictive grouping): performance remains similar or slightly improved on some datasets, indicating that Dr. MAMR is stable when grouping is more conservative. (2) 0.8 (more aggressive grouping): performance drops slightly, suggesting that overly broad grouping merges semantically distinct segments, which weakens the ability to provide stable CI. Overall, Dr. MAMR maintains strong performance for a wide range of thresholds, demonstrating robustness, but extremely permissive grouping can blur important distinctions in the reasoning trajectory.
>
> - (ii) **Effect of embedding model used for grouping**: By default, semantic grouping uses Qwen2.5-0.5B embeddings. To test robustness to this choice, we replace the embeddings with: (1) Qwen2-0.5B, a smaller instruction-tuned model (2) Qwen3-Embedding-0.6B, a specialized embedding model. As shown in Table below, the method remains stable across embedding variants. Notably, Qwen3-Embedding-0.6B improves grouping quality (e.g., Gaokao2023en: 65.20 → 66.49, Olympiad: 52.34 → 54.52), likely because its representations better capture semantic similarity. These results indicate that CI is robust to embedding model choice, and improvements can be obtained with stronger embedding encoders.
>
>
> | Variant                           | AIME24 | AMC23 | Gaokao2023en | Olympiad Bench |
> |----------------------------------|--------|--------|----------------|------------------|
> | Dr.MAMR                          | 20.00  | 62.50  | 65.20          | 52.34           |
> | w/ threshold 0.95 for CI         | 20.00  | 60.00  | 65.71          | 52.73           |
> | w/ threshold 0.8 for CI          | 16.67  | 57.50  | 62.86          | 50.20           |
> | w/ Qwen2-0.5B for CI             | 16.67  | 60.00  | 63.12          | 51.56           |
> | w/ Qwen3-Embedding-0.6B for CI   | 20.00  | 62.50  | 66.49          | 54.52           |

---

> ### Author Response · Authors · 2025-11-23
> **Response to W3 (2)**
>
> >### **(2) ablation study on $\alpha$ and $\beta$.**
>
> We thank the reviewer for raising this point. Following the experimental setup in Sec. 6.1, we conduct two sets of experiments: (1) fixing $\alpha = 0.1$ and varying $\beta \in {0.1, 0.3, 0.5}$, and (2) fixing $\beta = 0.1$ and varying $\alpha \in {0.1, 0.3, 0.5}$. The results are shown in the tables below.
>
> | Benchmark        | ReMA  | Dr. MAMR (β = 0.1) | (β = 0.3) | (β = 0.5) |
> |------------------|--------|---------------------|-----------|-----------|
> | MATH500          | 74.40  | 78.60              | 78.80     | 77.20     |
> | GSM8K            | 90.60  | 92.12              | 91.59     | 86.05     |
> | AIME24           | 13.33  | 20.00              | 20.00     | 16.67     |
> | AMC23            | 50.00  | 62.50              | 60.00     | 52.50     |
> | Gaokao2023en     | 57.92  | 65.20              | 66.50     | 63.12     |
> | Minerva Math     | 34.93  | 38.24              | 38.60     | 34.56     |
> | Olympiad Bench   | 42.58  | 52.34              | 51.60     | 45.04     |
> | **Average**      | 51.97  | **58.43**          | 58.16     | 53.15     |
>
>
> | Benchmark        | ReMA  | Dr. MAMR (α = 0.1) | (α = 0.3) | (α = 0.5) |
> |------------------|--------|---------------------|-----------|-----------|
> | MATH500          | 74.40  | 78.60              | 77.80     | 76.80     |
> | GSM8K            | 90.60  | 92.12              | 91.18     | 88.10     |
> | AIME24           | 13.33  | 20.00              | 16.67     | 13.33     |
> | AMC23            | 50.00  | 62.50              | 57.50     | 52.50     |
> | Gaokao2023en     | 57.92  | 65.20              | 64.16     | 62.08     |
> | Minerva Math     | 34.93  | 38.24              | 36.77     | 33.82     |
> | Olympiad Bench   | 42.58  | 52.34              | 50.00     | 42.77     |
> | **Average**      | 51.97  | **58.43**          | 56.30     | 52.77     |
>
>
> From the tables, we observe:
>
> - **Effect of varying $\beta$ (restart-behavior reward)**: When $\beta$ increases from 0.1 to 0.3, Dr. MAMR maintains very stable average performance (58.43 → 58.16). Even when $\beta$ is further increased to 0.5, where performance drops to 53.15, Dr.~MAMR still outperforms ReMA (51.97) by a clear margin.
> **Interpretation**: $\beta$ affects only the reasoning agent’s restart behavior.
> A moderate value of $\beta$ (0.1 to 0.3) guides this behavior without disrupting the main optimization process.
> However, an excessively large $\beta$ risks over-encouraging or over-penalizing restarts, which discourages beneficial exploratory steps and ultimately leads to degraded performance.
>
> - **Effect of varying $\alpha$ (causal-influence reward)**: Increasing $\alpha$ from 0.1 to 0.3 yields only a mild change in performance (58.43 to 56.30). Even at $\alpha = 0.5$, where the score is 52.77, Dr. MAMR maintains performance above the ReMA baseline (51.97), indicating that the method remains robust across a broad range of $\alpha$ values.
> **Interpretation** $\alpha$ encourages each agent to take actions that can influence the other’s internal reasoning trajectory. When $\alpha$ becomes large, this incentive may occasionally place more emphasis on cross-agent influence than on improving final reasoning quality. As a result, the overall performance tends to vary more noticeably with respect to $\alpha$ than to $\beta$.
>
> - **Overall robustness**: Across all combinations of $\alpha$ and $\beta$, mild hyperparameter choices (e.g., $\alpha = 0.1$ with $\beta = 0.3$, or $\beta = 0.1$ with $\alpha = 0.3$) achieve performance comparable to the best setting of $\alpha = 0.1, \beta = 0.1$. Even when the hyperparameters are set to larger values such as 0.5, Dr. MAMR still surpasses ReMA, as Dr.MAMR addresses the fundamental lazy-agent issue. Overall, these results show that Dr. MAMR provides robust improvements without requiring extensive hyperparameter tuning.

---

> ### Author Response · Authors · 2025-11-23
> **Response to W4 & Q1**
>
> >### W4: Connection to MARL: Comparative context to prior “lazy‑agent” literature could be tighter. The paper cites MARL work on lazy agents but does not empirically compare to mechanisms from that line (e.g., influence‑based credit or external‑state influence). This limits continuity with prior method. (Sec. 2, p. 3; cf. ICML’23 “Lazy Agents” & Lazy‑MDPs)
>
>
>
> We thank the reviewer for raising this point. While the MARL literature has indeed studied “lazy-agent’’ behaviors, the classical mechanisms developed there—such as influence-based credit assignment or external-state influence—do not transfer cleanly to LLM multi-agent reasoning.
>
>
>
> The core difficulty is that MARL counterfactual methods assume a small, discrete action space, where one can enumerate or approximate how an agent’s alternative action would change the joint outcome. In LLM multi-agent reasoning, however, an “action’’ is an entire natural-language message, and even a single-token modification can drastically alter the downstream reasoning of the other agent. Constructing semantically plausible counterfactual actions is therefore non-trivial, and evaluating their impact becomes intractable.
>
>
>
> In addition, previous MARL counterfactual-influence methods aim to measure how a different action would have changed another agent’s action. Our objective is fundamentally different: we quantify how the agent’s actual message contributes to the current intermediate state of the reasoning trajectory.
>
>
>
> Thus, while the high-level phenomenon is related, existing MARL lazy-agent solutions cannot be directly applied to multi-agent LLM reasoning. We appreciate the reviewer for highlighting this connection and will incorporate this clarification into the related-work discussion.
>
>   --------
>
>
> >### Q1: Theory stress tests: Beyond removing 1/T, did you try alternatives (e.g., adaptive normalization by effective turns that exclude no‑ops, or per‑role normalization)? Do the theoretical predictions hold when rewards differ slightly (near‑equal) or when token‑level normalizations are active? (Sec. 5.1, p. 5)
>
>
>
> >### **(1) alternative normalization methods**
>
>
>
> We appreciate the reviewer’s suggestions regarding alternative normalization schemes. However, we clarify that these approaches do not remove the underlying structural bias toward shorter trajectories.
>
>
>
> A key empirical observation (Fig. 5) already reveals the fundamental issue: trajectories exhibiting lazy-agent behavior are typically shorter. This is intuitive—once an agent becomes lazy, it stops contributing and the interaction terminates earlier. Under any length-based normalization, these shorter trajectories inevitably receive larger gradient weights simply because their denominators are smaller.
>
>
>
> - **Effective-turn normalization**. Removing no-op or blank turns does not alleviate this asymmetry. Lazy trajectories typically contain more low-content turns; filtering them out makes their effective length even shorter, whereas well-reasoned trajectories retain most turns. As a result, effective-turn normalization may actually amplify the weight advantage of lazy trajectories.
>
>
>
> - **Per-role normalization**. Normalizing per agent simply rescales contributions but does not change the ordering: lazy trajectories still give each agent fewer turns and therefore a smaller denominator. Well-reasoned trajectories remain down-weighted because their cooperative exchange inherently requires more turns.
>
>
>
> Overall, both proposed variants (effective-turn normalization and per-role normalization) retain the core property that shorter trajectories receive larger normalized gradients, meaning the bias identified in Theorem 1 remains intact. The only reliable way to eliminate this structural preference is to remove the turn-length normalization entirely, as done in our corrected objective.
>
>
>
> >### **(2)Do the theoretical predictions hold when rewards differ slightly (near‑equal) or when token‑level normalizations are active?**
>
>
>
> As we mentioned before, the presence of reward noise or token-level clipping does not alter the existence or direction of the bias. Such noise may make the magnitude of the bias less precisely measurable, but it does not reverse the fact that, under any length-based normalization, shorter trajectories are systematically up-weighted.

---

> ### Author Response · Authors · 2025-11-23
> **Response to Q2**
>
> >### Q2: The phenomenon of “lazy agents” has been studied in MARL (e.g., LAIES at ICML’23; Lazy‑MDPs). This paper’s contribution is to diagnose and mitigate it in multi‑agent LLM reasoning with a targeted loss fix, a tractable step‑level influence signal, and a restart‑credit mechanism. What's your next plan for further work to combine this work?
>
>
>
> We sincerely thank the reviewer for raising this insightful question and for the opportunity to discuss future directions.
>
>
>
> While prior MARL work such as LAIES and Lazy-MDPs has analyzed lazy-agent behavior in discrete-action Markov games, these frameworks rely on counterfactual deviations in small action spaces and therefore do not directly apply to high-dimensional language actions. Nevertheless, their underlying principles—counterfactual influence, credit assignment, and game-theoretic characterizations of laziness—are conceptually related to the phenomena we study in LLM-based multi-agent reasoning.
> A promising direction for future work is to bridge these theoretical foundations with the emerging LLM multi-agent setting. In particular, we plan to:
> Formalize an abstract game model where each agent’s “action” corresponds to a segment of natural-language reasoning, enabling a principled comparison between classical MARL credit-assignment theory and LLM reasoning dynamics.
> Extend influence-based and lazy-agent analyses to the sequential, message-passing interactions characteristic of LLM multi-agent systems, where even minor text changes can reshape downstream reasoning.
> This connection between classical MARL theory and LLM multi-agent reasoning offers a natural and exciting path forward, and we believe our current work provides the foundation for such broader theoretical development.

---

> > ### Comment · Reviewer_7UcQ · 2025-11-28
> > **Answer acknowledged**
> >
> > Thank the authors for their comprehensive and high-quality response. The rebuttal has effectively addressed my primary concerns regarding domain generality, method sensitivity, and theoretical assumptions.

---

> ### Author Response · Authors · 2025-11-28
> **Thank you for your support**
>
> Dear Reviewer 7UcQ,
>
> Thank you sincerely for your considerate follow-up and for noting that **our rebuttal has effectively addressed your concerns**. We truly appreciate the time and thoughtful attention you invested in evaluating our work.
>
> Thank you again for your constructive feedback and generous engagement throughout the review process!
>
> Best,
>
> Submission #14685 Authors

---

### Official Review · Reviewer_UntM · 2025-11-01

**Soundness:** 3
**Presentation:** 3
**Contribution:** 3
**Rating:** 6
**Confidence:** 3

**Summary:**

The paper studies lazy agent behavior in multi-agent LLM reasoning, where one agent does most of the work and the other contributes little. It attributes the collapse to a turn-averaging term in multi-turn GRPO that implicitly favors shorter dialogues. Their method Dr. MAMR removes the turn normalization, estimates step influence with a Shapley-style average over semantically similar steps across rollouts, and equips the reasoner with a restart token rewarded only when restarting improves the verifier-aligned likelihood of the final step. On math benchmarks with Qwen2.5 models at 3B, 7B, and 14B, the approach outperforms GRPO, VRP prompting, and ReMA, and shows better training stability and pass-at-K scaling. Ablations indicate that normalization removal and the influence estimator are key contributors.

**Strengths:**

- Clear diagnosis of the lazy agent failure mode with a simple insight into how turn normalization biases learning toward short dialogues.
- Method that is principled and modular with debiased objective, a practical influence estimator that reduces single-trajectory and phrasing bias, and a verifier-aligned restart mechanism.
- Consistent gains over strong baselines across multiple model sizes and math benchmarks, with improved stability and better pass-at-K behavior.
- Useful ablations and training dynamics analyses that support the design choices.

**Weaknesses:**

- Empirical scope restricted to math reasoning; claims about multia-gent benefits would be stronger with code and other domains requiring reasoning.
- Since the paper emphasizes single vs multi-agent comparison, it would be good to include compute vs performance for single agents and DR MAMR.

**Questions:**

A threshold of 0.9 was used for grouping, have you tried to vary this value and see how it affect the learning outcome of DR MAMR?

---

> ### Author Response · Authors · 2025-11-23
> **Response to W1 & W2**
>
> We sincerenly appreciate the reviewer’s recognition of our clear theoretical diagnosis, principled methodological design, and consistent empirical improvements. We address each of the reviewer’s concern in detail below.
>
> ---------
>
> >### W1: Empirical scope restricted to math reasoning;
>
> We thank the reviewer for raising this important point. To evaluate the generalization capability of Dr. MAMR beyond math reasoning, we conduct experiments on code generation benchmarks. We use the DeepCoder dataset for training and adopt DeepSeek-R1-Distill-Qwen-1.5B as the base model. We compare Dr. MAMR against two baselines: the single-agent GRPO and the multi-agent ReMA, evaluating performance on three widely used code reasoning benchmarks: LiveCodeBench, Codeforces, and HumanEval+. For each benchmark, we report pass@k metrics at $k \in {1, 2, 4, 8}$. More details and prompt can be found in H.5 in Appendix. We report results in table below:
>
>
> |Benchmark| Metric  | GRPO | ReMA | Dr. MAMR |
> |----|---------|------|------|-----------|
> |Codeforces| pass@1  | 5.88 | 4.66 | 6.22  |
> || pass@2  | 8.33 | 7.35 | 8.33  |
> || pass@4  | 11.03| 11.27| 12.25 |
> || pass@8  | 15.44| 14.95| 15.93 |
>
>
>
> |Benchmark| Metric  | GRPO  | ReMA  | Dr. MAMR |
> |----|---------|-------|-------|-----------|
> |HumanEval+| pass@1  | 58.28 | 57.67 | 61.35 |
> || pass@2  | 69.33 | 70.55 | 71.78 |
> || pass@4  | 77.30 | 76.69 | 79.14 |
> || pass@8  | 83.44 | 82.82 | 84.66 |
>
>
>
> |Benchmark| Metric  | GRPO  | ReMA  | Dr. MAMR |
> |-----|---------|-------|-------|-----------|
> |LiveCodeBench| pass@1  | 15.77 | 15.05 | 19.35 |
> || pass@2  | 19.35 | 17.56 | 22.58 |
> || pass@4  | 25.45 | 22.22 | 26.88 |
> || pass@8  | 29.39 | 24.37 | 32.26 |
>
>
>
> As shown in above tables, Dr. MAMR consistently outperforms both GRPO and ReMA across all benchmarks and almost all pass@$k$ settings. On LiveCodeBench, Dr. MAMR achieves a substantial improvement in pass@1 (19.35\%) compared to GRPO (15.77\%) and ReMA (15.05\%). Similar gains are observed on HumanEval+ (61.35\% vs. 58.28\%/57.67\%). These results demonstrate that Dr. MAMR not only excels in math reasoning but also generalizes effectively to complex code-based reasoning tasks, validating its robustness across diverse domains.
>
>
>
> **We further visualize the training dynamics of Dr. MAMR compared to ReMA in Figure 8 in the appendix**. The plotted curve reports the mean reward throughout the RL training process. We observe that while ReMA’s performance quickly plateaus and deteriorates—ultimately collapsing to near-zero values—Dr. MAMR demonstrates sustained and progressively improving performance. This indicates that Dr. MAMR enables stable optimization in long-horizon multi-agent reasoning tasks.
>
> -----------
>
> >### W2: Since the paper emphasizes single vs multi-agent comparison, it would be good to include compute vs performance for single agents and DR MAMR.
>
> We sincerely thank the reviewer for this helpful suggestion regarding compute–performance comparison.
>
> **(1) Multi-agent overhead is intrinsic and minimal in our setting.**
> In Dr. MAMR, the meta-thinking agent and reasoning agent share the same backbone LLM and differ only in system prompts. Thus, multi-agent rollouts do not require loading or running two separate models. The only additional cost is passing the updated conversation state between turns—an overhead inherent to _all_ multi-agent LLM systems rather than specific to Dr. MAMR, and empirically very small.
>
> **(2) CI estimation and restart reward incur marginal cost.**
> Both are computed on top of already generated trajectories, so the added overhead is marginal compared to the dominant cost of RL training (sampling multiple rollouts and performing GRPO updates).
>
> **(3) Compute comparison.**
> **To fully address your concern, in Fig. 9 (Appendix) we report the per-step training time of Dr. MAMR versus single-agent GRPO**. While Dr. MAMR is slightly slower, the difference is small relative to overall RL cost and is far outweighed by the substantial performance improvements it brings.

---

> ### Author Response · Authors · 2025-11-23
> **Response to Q1**
>
> >### Q1: A threshold of 0.9 was used for grouping, have you tried to vary this value and see how it affect the learning outcome of DR MAMR?
>
> We thank the reviewer for this insightful question. To address your concern, we provide an ablation study on threshold for CI estimator. Dr. MAMR uses a cosine-similarity threshold of 0.9 to decide whether two reasoning segments belong to the same group. To evaluate sensitivity to this design choice, we vary the threshold to 0.95 and 0.8. Results in Table below show: (1) 0.95 (more restrictive grouping): performance remains similar or slightly improved on some datasets, indicating that Dr. MAMR is stable when grouping is more conservative. (2) 0.8 (more aggressive grouping): performance drops slightly, suggesting that overly broad grouping merges semantically distinct segments, which weakens the ability to provide stable CI. Overall, Dr. MAMR maintains strong performance for a wide range of thresholds, demonstrating robustness, but extremely permissive grouping can blur important distinctions in the reasoning trajectory.
>
> | Variant                           | AIME24 | AMC23 | Gaokao2023en | Olympiad Bench |
> |----------------------------------|--------|--------|----------------|------------------|
> | Dr.MAMR                          | 20.00  | 62.50  | 65.20          | 52.34           |
> | w/ threshold 0.95 for CI         | 20.00  | 60.00  | 65.71          | 52.73           |
> | w/ threshold 0.8 for CI          | 16.67  | 57.50  | 62.86          | 50.20           |

---

### Official Review · Reviewer_s5AW · 2025-11-11

**Soundness:** 2
**Presentation:** 2
**Contribution:** 2
**Rating:** 4
**Confidence:** 3

**Summary:**

This paper examines a multi-agent LLM reasoning framework, ReMA, and finds that it induces lazy agent behavior (where one agent contributes little to the final result). The paper examines why this lazy agent behavior occurs, and introduces a verfiable reward mechanism that allows the agent to discard previous outputs and restart reasoning. This setup improves multi-agent reasoning behavior over the original ReMA on standard math benchmarks, and shows improved performance compared to single-agent frameworks.

**Strengths:**

* The problem is clearly identified and the theoretical analysis gives some insight into why this occurs (though it is an interesting choice to call this Dr. MAMR if the paper goes out of its way to claim that this is distinct from Dr. GRPO).

* The introduced causaul influence mechanism is a nice way to measure the impact of the restart action in a computationally-efficient manner (though the effectiveness of the proposed semantic similarity is not really analyzed).

* The results show pretty clear improvements over the original ReMA.

**Weaknesses:**

* My biggest concern is that the lazy agent behavior (and corresponding proposed solution) is only applicable to one recent multi-agent framework, ReMA. Given that this is a recent (and to this point not very widely used) framework with relatively poor performance (it under-performs single-agent GRPO), it's unclear whether this problem/solution will have much impact. For example, there's nothing to indicate that other multi-agent LLM frameworks will induce similar behavior. The fact that no other multi-agent LLM frameworks are included as baselines do not give me much confidence here.

* Experimental results are only over one model family (Qwen 2.5) and one type of benchmark (math).

* There's no sensitivity analysis on the new additional hyperparameters, e.g. alpha/beta. There's no analysis on whether there's a risk of potentially entering modes where restarts are over-triggered, for example.

**Questions:**

1. Fig. 3 is also not clear. What is the "performance gap" that is being measured here? Is this ReMA+ val. acc - ReMA val. acc? Why does the gap increase between Pass@1 and Pass@16 for AIME25/AIME24?

2. I don't understand Fig. 2. Particularly the difference from (b) to (c) to (d). What is the "causal-effect gap" here? A higher causal influence is better, yes? But it doesn't seem like there is any significant difference in the reasoning agent's causal influence from (b) to (d), for example.

---

> ### Author Response · Authors · 2025-11-23
>
> We thank the reviewer for the thoughtful assessment of our work and for the constructive comments. Below, we address each of the reviewer’s concerns point-by-point.
>
> --------
>
> >### W1: The lazy agent behavior and corresponding solution seem specific to ReMA—a recent, not widely adopted framework that underperforms the single-agent GRPO baseline. Without evidence that similar issues occur in other multi-agent LLM frameworks, and given the absence of broader baseline comparisons
>
> >### **(1) lazy agent behavior arises from a _general structural property_ of multi-agent RL systems when rewards are assigned only on the final outcome**
>
> We thank the reviewer for raising this concern. We respectfully clarify that the lazy-agent phenomenon is _not_ specific to ReMA, but arises from a _general structural property_ of multi-agent RL systems when rewards are assigned only on the final outcome.
>
> - **(i) Lazy-agent collapse is a well-known failure mode in multi-agent RL literature**: Prior work in multi-agent RL has documented that when multiple agents share a _single global reward_, individual agents tend to free-ride on the others [1][2], leading to _lazy_ or _inactive_ policies. The core reason is: **credit assignment is ambiguous when every agent receives the same final reward.**
>
> - **(ii) The issue is not specific to ReMA — it arises whenever multi-agent systems use R1-style training**: The lazy-agent behavior does not stem from the ReMA architecture itself. Instead, it arises from the **reward structure induced by R1-based RL**, which has become _widely adopted and empirically effective_ in reasoning-oriented RL following DeepSeek-R1. Because only the _final_ answer is verifiable, every agent in a multi-agent system receives **the same terminal reward**, regardless of its individual contribution. This inherently makes credit assignment ambiguous and naturally encourages laziness.
>
> - **This mechanism is shared by any multi-agent LLM system trained with R1-style rewards**, whether the interaction pattern is planner–solver, supervisor–worker, debate, or CoT+verifier. The architectural details differ, but the credit-assignment limitation is identical whenever training relies on a single verifiable final reward.
>
> ---------
>
> >### **(2) our solution is framework-agnostic and plug-and-play for a broad family of multi-agent LLM systems**
>
> We appreciate the reviewer’s concern. However, our contributions are **not tied to the ReMA architecture** and are intentionally designed to be **framework-agnostic** and **plug-and-play** for a broad family of multi-agent LLM systems.
>
> - **(i) the training objective used by ReMA—multi-turn GRPO—is _not_ unique to ReMA**. Multi-turn GRPO and its variants have recently been adopted widely in LLM RL, including multi-agent settings and multi-turn single-agent RL [3][4][5][6][7]. Our theoretical analysis identifies a fundamental bias in the turn-normalization term that inherently favors shorter trajectories over longer but equally rewarded ones. This insight is applicable to **any** framework built upon multi-turn GRPO, and provides guidance for future multi-agent and multi-turn RL designs beyond ReMA itself.
>
> - **(ii) our Shapley-inspired causal-influence estimator is not tied to the ReMA architecture, nor does it _require_ any specialized multi-agent structure.** Rather, it can be **naturally adapted** to the widely adopted interaction pattern in modern LLM multi-agent systems, where agents communicate **sequentially** and jointly produce a single final answer.
>
> - **(iii) we propose a novel restart deliberation mechanism that targets a fundamental failure mode in multi-agent LLM interactions**: agents becoming lost or drifting during prolonged multi-turn exchanges. This phenomenon has already been documented in LLM multi-turn human–model conversations, and our preliminary analysis in Fig. 3 further confirms that similar issue occurs in multi-agent settings as well. Importantly, this issue is **not specific to ReMA**. It can arise in _any_ multi-agent LLM framework where agents communicate over several turns. Our proposed restart behavior and its associated reward mechanism therefore serve as a **general mechanism** for stabilizing long-horizon multi-agent reasoning, helping agents recover from accumulated errors..
>
> In summary, although our experiments use ReMA as the testbed, our analysis and solutions are not specific to ReMA. The issues we identify, including turn bias, credit-assignment ambiguity, and multi-turn drift, arise broadly in RL-trained multi-agent LLM systems. Our proposed fixes are plug-and-play for any such framework, so the impact of our work extends well beyond ReMA.

---

> ### Author Response · Authors · 2025-11-23
> **Response to W1 (2)**
>
> >### **(3) ReMA’s rapid adoption and limitations make our contribution timely and impactful**
>
> We respectfully clarify that although ReMA is a recent framework, both its _research impact_ and its _underlying training paradigm_ make it **highly meaningful within the rapidly growing area of multi-agent RL and LLM reasoning**.
>
> - **(i) ReMA is not an obscure framework—its adoption and influence are growing rapidly:** ReMA was accepted to NeurIPS 2025, and despite being published only recently, it has already accumulated 32 citations within a few months. This indicates active research interest in multi-agent LLM reasoning and suggests that ReMA is serving as an early template for how such systems may be designed. **Far from being a niche prototype, ReMA has quickly become a point of reference for multi-agent LLM RL, and our work provides timely analysis of its limitations and potential.**
>
>
> - **(ii) The underlying training paradigm—multi-turn GRPO—is widely used beyond ReMA:** ReMA’s training objective is based on multi-turn GRPO, a paradigm that is being actively explored in several concurrent works [3][4][5][6][7] and applicable to any multi-agent or hierarchical reasoning system. Our theoretical analysis reveals an **inherent bias** in multi-turn GRPO’s turn-normalization, a phenomenon that impacts any framework built on this paradigm—not just ReMA.
> Thus, **our contribution provides insights that directly benefit the broader community working on multi-agent RL and multi-turn LLM reasoning**.
>
> - **(iii) ReMA’s performance does not limit the significance and impact of our contribution.** Although ReMA underperforms single-agent GRPO in our experiments, we believe this _highlights the importance_ of identifying and resolving multi-agent failures early. In fact, our results clearly show that **multi-agent LLM reasoning _can_ succeed—ReMA’s weaknesses stem from correctable design issues, not from the multi-agent paradigm itself.** Our contribution identifies why it underperforms and introduces **general, plug-and-play solutions**—causal-influence credit assignment, restart-reward stabilization, and a corrected multi-turn GRPO objective—that substantially boost performance. Thus, the contribution of our work is not in defending ReMA’s original performance, but in **demonstrating how to make multi-agent LLM genuinely effective in practice**, providing insights that extend well beyond a single framework.
>
> ------------------------
>
> >### **(4) ReMA is the only existing end-to-end RL framework for cooperative multi-agent LLM reasoning**
>
> At the time of submission, to our best knowledge, ReMA is the only work that applies end-to-end RL with verifiable final rewards (R1-style training) to cooperative multi-agent LLM reasoning on math benchmarks. However, as discussed above, the lazy-agent behavior is _not_ specific to ReMA. It naturally arises whenever R1-style RL is applied to a multi-agent framework where all agents receive the same terminal reward and individual credit becomes unidentifiable.
>
> A recent concurrent work SPIRAL [8] trains two agents via self-play in zero-sum games (e.g., Kuhn Poker), and uses math benchmarks only to test generalization. Its training setup and overall pipeline are fundamentally different from ours. Therefore, including SPIRAL as a baseline would not constitute a fair comparison. And our work remains among the first to analyse the structural failures of multi-agent RL for LLM reasoning and propose broadly applicable solutions.
>
> ----
>
> [1] Lazy agents: A new perspective on solving sparse reward problem in multi-agent reinforcement learning. ICML 2023.
>
> [2] Social Influence as Intrinsic Motivation for Multi-Agent Deep Reinforcement Learning. ICML 2019.
>
> [3] Webagent-r1: Training web agents via end-to-end multi-turn reinforcement learning.
>
> [4] MobileGUI-RL: Advancing Mobile GUI Agent through Reinforcement Learning in Online Environment.
>
> [5] Simpletir: End-to-end reinforcement learning for multi-turn tool-integrated reasoning
>
> [6] Not all thoughts are generated equal: Efficient llm reasoning via multi-turn reinforcement learning
>
> [7] Segment policy optimization: Effective segment-level credit assignment in rl for large language models.
>
> [8] SPIRAL: Self-Play on Zero-Sum Games Incentivizes Reasoning via Multi-Agent Multi-Turn Reinforcement Learning

---

> ### Author Response · Authors · 2025-11-23
> **Response to W2**
>
> >### W2: Experimental results are only over one model family (Qwen 2.5) and one type of benchmark (math).
>
>
> >### **(1) results on different model family**
>
> We thank the reviewer for pointing this out. Following the experimental setup described in Sec. 6.1, we conduct experiments using two base models: Meta-Llama-3-8B-Instruct and Llama-3.1-8B-Instruct. We compare our proposed Dr. MAMR framework against ReMA and the single-agent GRPO baseline. The results are summarized in Table below.
>
> |Base Model| Benchmark         | GRPO  | ReMA  | Dr. MAMR |
> |-----|-------------------|--------|--------|----------|
> |Llama-3-8B-Instruct| MATH500           | 40.40  | 33.80  | 43.20    |
> || GSM8K             | 82.79  | 79.38  | 83.47    |
> || AIME24            | 0.00   | 0.00   | 3.33     |
> || Gaokao2023en      | 29.87  | 28.57  | 32.49    |
> || Minerva Math      | 17.28  | 13.97  | 22.06    |
> || Olympiad Bench    | 10.52  | 8.89   | 16.30    |
> || **Average**       | 29.40  | 26.73  | 33.34    |
>
>
> |Base Model| Benchmark         | GRPO  | ReMA  | Dr. MAMR |
> |-----|-------------------|--------|--------|----------|
> |Llama-3.1-8B-Instruct| MATH500           | 55.20  | 53.20  | 58.60    |
> || GSM8K             | 88.32  | 87.26  | 90.22    |
> || AIME24            | 10.00  | 6.67   | 13.33    |
> || AMC23             | 27.50  | 20.00  | 30.50    |
> || Gaokao2023en      | 44.94  | 37.14  | 48.95    |
> || Minerva Math      | 32.35  | 28.31  | 36.77    |
> || Olympiad Bench    | 16.44  | 19.56  | 22.67    |
> || **Average**       | 39.25  | 36.97  | 43.01    |
>
> Across both Llama base models—which overall exhibit lower raw performance than the Qwen2.5 series—we consistently observe that Dr. MAMR yields substantial improvements over both ReMA and GRPO. This demonstrates that the benefits of our multi-agent meta-reasoning framework persist even with weaker base models, highlighting its robustness and general applicability.
>
>
> >### **(2) results only on math benchmark**
>
> To evaluate the generalization capability of Dr. MAMR beyond math reasoning, we conduct experiments on code generation benchmarks. We use the DeepCoder dataset for training and adopt DeepSeek-R1-Distill-Qwen-1.5B as the base model. We compare Dr. MAMR against two baselines: the single-agent GRPO and the multi-agent ReMA, evaluating performance on three widely used code reasoning benchmarks: LiveCodeBench, Codeforces, and HumanEval+. For each benchmark, we report pass@k metrics at $k \in {1, 2, 4, 8}$. More details and prompt can be found in H.5 in Appendix. We report results in table below:
>
>
> |Benchmark| Metric  | GRPO | ReMA | Dr. MAMR |
> |----|---------|------|------|-----------|
> |Codeforces| pass@1  | 5.88 | 4.66 | 6.22      |
> || pass@2  | 8.33 | 7.35 | 8.33      |
> || pass@4  | 11.03| 11.27| 12.25     |
> || pass@8  | 15.44| 14.95| 15.93     |
>
> |Benchmark| Metric  | GRPO  | ReMA  | Dr. MAMR |
> |----|---------|-------|-------|-----------|
> |HumanEval+| pass@1  | 58.28 | 57.67 | 61.35     |
> || pass@2  | 69.33 | 70.55 | 71.78     |
> || pass@4  | 77.30 | 76.69 | 79.14     |
> || pass@8  | 83.44 | 82.82 | 84.66     |
>
> |Benchmark| Metric  | GRPO  | ReMA  | Dr. MAMR |
> |-----|---------|-------|-------|-----------|
> |LiveCodeBench| pass@1  | 15.77 | 15.05 | 19.35     |
> || pass@2  | 19.35 | 17.56 | 22.58     |
> || pass@4  | 25.45 | 22.22 | 26.88     |
> || pass@8  | 29.39 | 24.37 | 32.26     |
>
> As shown in above tables, Dr. MAMR consistently outperforms both GRPO and ReMA across all benchmarks and almost all pass@$k$ settings. On LiveCodeBench, Dr. MAMR achieves a substantial improvement in pass@1 (19.35\%) compared to GRPO (15.77\%) and ReMA (15.05\%). Similar gains are observed on HumanEval+ (61.35\% vs. 58.28\%/57.67\%). These results demonstrate that Dr. MAMR not only excels in math reasoning but also generalizes effectively to complex code-based reasoning tasks, validating its robustness across diverse domains.
>
> **We further visualize the training dynamics of Dr. MAMR compared to ReMA in Figure 8 in the appendix**. The plotted curve reports the mean reward throughout the RL training process. We observe that while ReMA’s performance quickly plateaus and deteriorates—ultimately collapsing to near-zero values—Dr. MAMR demonstrates sustained and progressively improving performance. This indicates that Dr. MAMR enables stable optimization in long-horizon multi-agent reasoning tasks.

---

> ### Author Response · Authors · 2025-11-23
> **Response to W3 (1)**
>
> >### W3: There's no sensitivity analysis on the new additional hyperparameters, e.g. alpha/beta. There's no analysis on whether there's a risk of potentially entering modes where restarts are over-triggered, for example.
>
> >### **(1) sensitivity analysis**
>
> We thank the reviewer for raising this point. Following the experimental setup in Sec. 6.1, we conduct two sets of experiments: (1) fixing $\alpha = 0.1$ and varying $\beta \in {0.1, 0.3, 0.5}$, and (2) fixing $\beta = 0.1$ and varying $\alpha \in {0.1, 0.3, 0.5}$. The results are shown in the tables below.
>
> | Benchmark        | ReMA  | Dr. MAMR (β = 0.1) | (β = 0.3) | (β = 0.5) |
> |------------------|--------|---------------------|-----------|-----------|
> | MATH500          | 74.40  | 78.60              | 78.80     | 77.20     |
> | GSM8K            | 90.60  | 92.12              | 91.59     | 86.05     |
> | AIME24           | 13.33  | 20.00              | 20.00     | 16.67     |
> | AMC23            | 50.00  | 62.50              | 60.00     | 52.50     |
> | Gaokao2023en     | 57.92  | 65.20              | 66.50     | 63.12     |
> | Minerva Math     | 34.93  | 38.24              | 38.60     | 34.56     |
> | Olympiad Bench   | 42.58  | 52.34              | 51.60     | 45.04     |
> | **Average**      | 51.97  | **58.43**          | 58.16     | 53.15     |
>
>
> | Benchmark        | ReMA  | Dr. MAMR (α = 0.1) | (α = 0.3) | (α = 0.5) |
> |------------------|--------|---------------------|-----------|-----------|
> | MATH500          | 74.40  | 78.60              | 77.80     | 76.80     |
> | GSM8K            | 90.60  | 92.12              | 91.18     | 88.10     |
> | AIME24           | 13.33  | 20.00              | 16.67     | 13.33     |
> | AMC23            | 50.00  | 62.50              | 57.50     | 52.50     |
> | Gaokao2023en     | 57.92  | 65.20              | 64.16     | 62.08     |
> | Minerva Math     | 34.93  | 38.24              | 36.77     | 33.82     |
> | Olympiad Bench   | 42.58  | 52.34              | 50.00     | 42.77     |
> | **Average**      | 51.97  | **58.43**          | 56.30     | 52.77     |
>
>
> From the tables, we observe:
>
> - **Effect of varying $\beta$ (restart-behavior reward)**: When $\beta$ increases from 0.1 to 0.3, Dr. MAMR maintains very stable average performance (58.43 → 58.16). Even when $\beta$ is further increased to 0.5, where performance drops to 53.15, Dr.~MAMR still outperforms ReMA (51.97) by a clear margin.
> **Interpretation**: $\beta$ affects only the reasoning agent’s restart behavior.
> A moderate value of $\beta$ (0.1 to 0.3) guides this behavior without disrupting the main optimization process.
> However, an excessively large $\beta$ risks over-encouraging or over-penalizing restarts, which discourages beneficial exploratory steps and ultimately leads to degraded performance.
>
> - **Effect of varying $\alpha$ (causal-influence reward)**: Increasing $\alpha$ from 0.1 to 0.3 yields only a mild change in performance (58.43 to 56.30). Even at $\alpha = 0.5$, where the score is 52.77, Dr. MAMR maintains performance above the ReMA baseline (51.97), indicating that the method remains robust across a broad range of $\alpha$ values.
> **Interpretation** $\alpha$ encourages each agent to take actions that can influence the other’s internal reasoning trajectory. When $\alpha$ becomes large, this incentive may occasionally place more emphasis on cross-agent influence than on improving final reasoning quality. As a result, the overall performance tends to vary more noticeably with respect to $\alpha$ than to $\beta$.
>
> - **Overall robustness**: Across all combinations of $\alpha$ and $\beta$, mild hyperparameter choices (e.g., $\alpha = 0.1$ with $\beta = 0.3$, or $\beta = 0.1$ with $\alpha = 0.3$) achieve performance comparable to the best setting of $\alpha = 0.1, \beta = 0.1$. Even when the hyperparameters are set to larger values such as 0.5, Dr. MAMR still surpasses ReMA, as Dr.MAMR addresses the fundamental lazy-agent issue. Overall, these results show that Dr. MAMR provides robust improvements without requiring extensive hyperparameter tuning.

---

> ### Author Response · Authors · 2025-11-23
> **Response to W3 (2)**
>
> >### **(2) whether there's a risk of potentially entering modes where restarts are over-triggered**
>
> We thank the reviewer for raising such insightful question. To address your concern, we provide a detailed analysis of the training dynamics associated with the restart behavior in Dr. MAMR. Following the experimental setting described in Sec. 6.1, we set $\alpha=\beta=0.1$, use 8 rollouts per prompt, and adopt a batch size of 128. At each global step, we track three quantities: (1) the mean reward that Dr. MAMR obtained on the training batch, (2) the total number of restart actions triggered by the reasoning agent, and (3) the average reward assigned to those restart actions. **The results are visualized in Fig. 7 in the appendix.**
>
> From the figure, several important patterns emerge. First, as training progresses, the overall mean reward of Dr. MAMR increases steadily, and the mean reward associated specifically with restart actions exhibits a similar upward trend. This indicates that the restart reward signal is informative: it provides consistent and meaningful feedback to the reasoning agent, rather than introducing noise or instability. The agent learns to exploit the restart mechanism in a way that aligns with improved task performance.
>
> Second, although the number of restart actions experiences a mild increase during the earliest phase of training—an expected effect due to the restart reward signal not yet being fully calibrated—this trend reverses quickly.
> After approximately 40 training steps, the number of restart actions begins to decline and eventually stabilizes around step 100. While the restart frequency does decrease, the reduction is only partial, indicating that the agent is learning to avoid unnecessary or redundant restarts. Importantly, this stabilization occurs despite the fact that the reward for restart actions continues to increase. In other words, the positive restart reward does not cause the agent to enter a degenerate mode where restarts are over-triggered simply to accumulate reward.
>
> **This observation is precisely the behavior we aim to achieve: the reasoning agent is not “hacked” into producing restart tokens blindly. Instead, it learns when a restart is genuinely beneficial based on the current prompt and the intermediate reasoning trajectory.**
>
> Together, these findings provide compelling evidence that the restart mechanism in Dr. MAMR is both stable and well-behaved. **Even when rewarded positively, the model does not collapse into overuse of restart behavior. Instead, it balances restart decisions in a principled manner, demonstrating robustness and reliability in practical training conditions.**

---

> ### Author Response · Authors · 2025-11-23
> **Response to Q1&Q2**
>
> >### Q1: Fig. 3 is also not clear. What is the "performance gap" that is being measured here? Why does the gap increase between Pass@1 and Pass@16 for AIME25/AIME24?
>
> We thank the reviewer for pointing out this ambiguity. We clarify that in Fig. 3, the “performance gap” is defined as the difference in validation accuracy between ReMA⁺ and the original ReMA, i.e., $Gap=ValAcc(ReMA+)−ValAcc(ReMA).$
> Although we stated in L320 that ReMA⁺ outperforms ReMA, we apologize for not explicitly defining this quantity in the figure.
>
> Regarding Pass@1 vs. Pass@16, these are distinct evaluation metrics:
> -   Pass@1 checks whether the model solves the problem in a single attempt.
> -   Pass@16 checks whether the model solves the problem using up to 16 attempts, which provides additional opportunities for exploration and recovery from early mistakes.
>
> And we observe that the performance gap between ReMA⁺ and ReMA increases under **Pass@16**, especially on AIME24/AIME25. Our interpretation is as follows:
> -   **ReMA⁺ explicitly incentivizes the `<restart>` behavior**, allowing the reasoning agent to abandon unproductive or incorrect partial chains and explore alternative solution paths.
> -   **ReMA, in contrast, tends to continue reasoning within a noisy or corrupted context once it becomes “lost,”** which limits its ability to recover even when multiple attempts are allowed.
>
> Thus, when more attempts are permitted (Pass@16), ReMA⁺ benefits more from its improved exploration ability, leading to a larger performance advantage over ReMA. This trend is also consistent with our scaling results in Fig. 9, where the performance gap between Dr. MAMR and ReMA **widens as Pass@K increases** on the more difficult benchmarks.
>
>
> >###  Q2: I don't understand Fig. 2. Particularly the difference from (b) to (c) to (d). What is the "causal-effect gap" here? A higher causal influence is better, yes? But it doesn't seem like there is any significant difference in the reasoning agent's causal influence from (b) to (d), for example.
>
> >### **(1) What is the "causal-effect gap" here? A higher causal influence is better, yes?**
>
> We thank the reviewer for this question and apologize for the lack of clarity. **The “causal-effect gap” refers to the difference between the causal influence of the meta-thinking agent and the causal influence of the reasoning agent.** Formally, it measures how imbalanced the two agents’ contributions are.
>
> Importantly, **a higher causal influence is _not_ necessarily better.** As stated in L208 of the paper, the key problem in ReMA is that _the reasoning agent’s causal influence collapses_, while the meta-thinking agent dominates. **What we want is balanced contributions, not simply a higher value.**
>
> >### **(2) Why (b) and (d) seem similar for the reasoning agent?**
>
> In panel **(b)** (ReMA), we intentionally selected the _best performing checkpoint_—the point before the reasoning agent fully collapses into a lazy agent. Even at this best moment, the reasoning agent’s causal influence is much lower than the meta-thinking agent’s, revealing the persistent imbalance.
>
> In contrast, in panel **(d)** (Dr. MAMR), although the reasoning agent’s absolute causal influence may appear numerically similar to that in (b), its relative contribution compared to the meta-thinking agent is much more balanced. This is the central improvement that our causal-influence module brings.
>
> This distinction becomes clearer when observing training dynamics: **As shown in Fig. 4**, under standard ReMA training, the reasoning agent’s causal influence gradually collapses toward zero as training continues—demonstrating the lazy-agent failure mode. Our method prevents this collapse and keeps both agents contributing meaningfully.

---

> ### Author Response · Authors · 2025-11-23
> **Response to minor points mentioned in  Strengths**
>
> >### though it is an interesting choice to call this Dr. MAMR if the paper goes out of its way to claim that this is distinct from Dr. GRPO
>
>  We thank the reviewer for the comment. We name our method Dr. MAMR to highlight the central message of our paper: **multi-agent RL _can succeed_ when designed correctly** (“done right”). The name is chosen to emphasize this conceptual contribution, not to imply any connection to Dr. GRPO.
>
>
> >### the effectiveness of the proposed semantic similarity is not really analyzed
>
> To address your concern, we conduct an ablation study comparing **Dr. MAMR** with its **w/o Shapley in CI** variant, which removes only the Shapley-based semantic-similarity grouping from the causal-influence module. As shown in the table below, removing Shapley grouping worsens performance across all benchmarks (e.g., AIME24: 20.00 → 16.67, AMC23: 62.50 → 60.00, Olympiad: 52.34 → 49.22). This confirms that the Shapley mechanism meaningfully contributes to the stability and effectiveness of the causal-influence reward.
>
> | Variant                           | AIME24 | AMC23 | Gaokao2023en | Olympiad Bench |
> |----------------------------------|--------|-------|---------------|----------------|
> | Dr.MAMR                          | 20.00  | 62.50 | 65.20         | 52.34          |
> | w/o CI                           | 13.33  | 52.50 | 63.38         | 45.31          |
> | w/o Shapley in CI                | 16.67  | 60.00 | 64.42         | 49.22          |

---

### Author Response · Authors · 2025-12-02
**Summary to Assist the Area Chair’s Evaluation (1)**

Dear Area Chair,

We sincerely thank the AC for their time and effort in evaluating our paper. Below, we provide a concise summary intended to assist the AC in assessing our work and understanding the key points raised and addressed during the rebuttal phase.

----------

>## **Summary of strengths mentioned by reviewers**

### **1. Clear motivation, strong theoretical analysis, and principled design**

-   **Reviewer 7UcQ:** “Crisp theoretical diagnosis of the cause. Theorem 1 formalizes how the 1/T normalization biases learning toward shorter trajectories, explaining the emergence of lazy agents.”

-   **Reviewer UntM:** “Clear diagnosis of the lazy-agent failure mode with a simple insight into how turn normalization biases learning toward short dialogues.”

-   **Reviewer eVXe:** “The proposed changes and additions to the loss and reward formulation are principled and experimentally backed with controlled ablations.”

-   **Reviewer s5AW:** “The problem is clearly identified and the theoretical analysis gives some insight into why this occurs.”


All reviewers agreed that the paper provides a _well-motivated and rigorous theoretical account_ of lazy-agent collapse in multi-agent LLM reasoning and offers a _principled fix_ grounded in reinforcement-learning theory.

----------

### **2. Novel methodological framework**


Reviewers consistently praised **Dr. MAMR** for its **novel and principled design**, emphasizing that the proposed components—the debiased objective, Shapley-style causal-influence estimator, and restart mechanism—are **interesting, computationally efficient, and well-structured**.

-   **Reviewer 7UcQ:** “Method that is principled and modular with a debiased objective, a practical influence estimator that reduces phrasing bias and variance, and a verifier-aligned restart mechanism.”

-   **Reviewer UntM:** “Principled and modular method with a practical influence estimator and verifier-aligned restart mechanism.”

-   **Reviewer eVXe:** “Shapley values are an interesting and, to my knowledge, novel way to provide dense credit attribution in multi-turn settings with relatively small computational overhead.”

-   **Reviewer s5AW:** “The introduced causal-influence mechanism is a nice way to measure the impact of the restart action in a computationally efficient manner.”

----------

### **3. Comprehensive experiments with consistent and generalizable gains**

-   **Reviewer 7UcQ:** “On seven math benchmarks with Qwen2.5 7B and 14B, Dr. MAMR outperforms GRPO, VRP, and ReMA, and improves stability.”

-   **Reviewer UntM:** “Consistent gains over strong baselines across multiple model sizes and math benchmarks, with improved stability and better pass-at-K behavior.”

-   **Reviewer eVXe:** “It’s great to see Dr. MAMR brings gains over other algorithmic changes, and that gains generalize to other domains and models.”

-   **Reviewer s5AW:** “The results show clear improvements over the original ReMA.”


During the rebuttal, we further added **cross-domain experiments (code reasoning)** and **multi-model evaluations (Llama 3-8B / 3.1-8B)**. All reviewers who engaged with these updates acknowledged that our results demonstrate strong and _robust generalization beyond math reasoning_.

---

> ### Author Response · Authors · 2025-12-02
> **Summary to Assist the Area Chair’s Evaluation (2)**
>
> >## **Reviewers’ follow-ups acknowledged effective rebuttal and improved assessments**
>
> Two reviewers explicitly confirmed that our clarifications and new experiments fully addressed their earlier concerns. Before the discussion freeze, the paper’s scores were **(8, 6, 6, 4)** with an **average of 6.0**.
>
> -   **Reviewer eVXe:** “My concerns have been largely addressed. It’s great to see Dr. MAMR brings gains over other algorithmic changes … I’ll update my score.” **Reviewer eVXe raised the score from 6 → 8 prior to the discussion freeze.**
>
> -   **Reviewer 7UcQ:** “Thank the authors for their comprehensive and high-quality response. The rebuttal has effectively addressed my primary concerns regarding domain generality, method sensitivity, and theoretical assumptions.” **Reviewer 7UcQ gave a positive initial rating (score 6) and stated that all concerns were fully addressed after the rebuttal**. However, by the time of their follow-up response, the review editing window had already closed, leaving no opportunity to update the score.
>
> -   **Reviewer UntM** provided a **positive initial review (score 6)** highlighting our clear theoretical diagnosis, modular method design, and consistent improvements. We submitted extensive new experiments on code reasoning and compute–performance comparisons to address their suggestions, though the reviewer did not provide a follow-up comment before the discussion freeze.
>
> - **Reviewer s5AW** initially gave a score of 4, primarily concerned that the lazy-agent issue might be limited to the ReMA framework and that our proposed solutions might lack generality. In our rebuttal, we provided detailed clarifications and extensive supporting experiments demonstrating that **(i) the lazy-agent behavior arises from a general structural property of R1-style multi-agent RL, rather than any ReMA-specific design choices**—a point **explicitly acknowledged by Reviewer 7UcQ**, and **(ii) our proposed fixes (causal-influence estimation, restart reward, and normalization correction) are framework-agnostic and plug-and-play across various multi-agent LLM system design**, which was **also recognized by Reviewer UntM and 7UcQ**. These clarifications directly addressed the reviewer’s concern; however, **the reviewer did not respond during the discussion phase**.
>
> **In summary:**  The two reviewers who engaged in discussion explicitly acknowledged that all their concerns had been resolved. The overall review consensus recognizes the paper as **theoretically well-founded, methodologically innovative, empirically validated, and broadly impactful** within the emerging area of multi-agent LLM reinforcement learning.
>
> ----------
>
> We hope this summary assists the Area Chair in conveniently and effectively evaluating our paper.
> ### **Once again, we would like to express our deepest gratitude for your time, effort, and thoughtful consideration devoted to our work.**

---

### Meta-Review · Area_Chair_iog3 · 2025-12-26

**Summary:**

In this paper, the proposed framework successfully mitigates this collapse through a debiased training objective, a Shapley-inspired causal influence estimator for dense credit assignment, and a verified restart mechanism that stabilizes long-horizon reasoning. Empirical evaluations confirm that the method consistently outperforms strong baselines including ReMA, standard GRPO, and advanced variants like DAPO across multiple model families and scales.

**Reviewer Concerns:**

The rebuttal successfully addressed concerns regarding limited empirical scope and weak baselines by extending evaluations to code benchmarks, Llama models, and advanced optimization variants. Issues related to hyperparameter sensitivity, computational overhead, and training stability were resolved through new ablation studies and dynamic analyses. The theoretical validity of the lazy agent diagnosis was effectively clarified, leaving no significant outstanding concerns.

**Reviewer Scores:**

Reviewer s5AW would likely raise the rating to meet the acceptance threshold. Reviewer eVXe is set to increase their score significantly with the addition of advanced baselines and cross-domain evaluations. The other reviewers would likely maintain their favorable positions.

---

### Decision · Program_Chairs · 2026-01-26

Accept (Poster)